# A Subunit of the COP9 Signalosome, MoCsn6, Is Involved in Fungal Development, Pathogenicity, and Autophagy in Rice Blast Fungus

Zi-Fang Shen,[a] Lin Li,[b] Jing-Yi Wang,[a] Yun-Ran Zhang,[a] Zi-He Wang,[a] Shuang Liang,[b] Xue-Ming Zhu,[b] ⓘ Jian-Ping Lu,[c] Fu-Cheng Lin,[a,b] ⓘ Xiao-Hong Liu[a]

[a]State Key Laboratory for Managing Biotic and Chemical Treats to the Quality and Safety of Agro-products, Institute of Biotechnology, Zhejiang University, Hangzhou, China
[b]State Key Laboratory for Managing Biotic and Chemical Treats to the Quality and Safety of Agro-products, Institute of Plant Protection and Microbiology, Zhejiang Academy of Agricultural Sciences, Hangzhou, China
[c]College of Life Sciences, Zhejiang University, Hangzhou, China

**ABSTRACT** The COP9 signalosome (CSN) is a highly conserved protein complex in eukaryotes, affecting various development and signaling processes. To date, the biological functions of the COP9 signalosome and its subunits have not been determined in *Magnaporthe oryzae*. In this study, we characterized the CSN in *M. oryzae* (which we named MoCsn6) and analyzed its biological functions. MoCsn6 is involved in fungal development, autophagy, and plant pathogenicity. Compared with the wild-type strain 70-15, Δ*Mocsn6* mutants showed a significantly reduced growth rate, sporulation rate, and germ tube germination rate. Pathogenicity assays showed that the Δ*Mocsn6* mutants did not cause or significantly reduced the number of disease spots on isolated barley leaves. After the *MoCSN6* gene was complemented into the Δ*Mocsn6* mutant, vegetative growth, sporulation, and pathogenicity were restored. The Osm1 and Pmk1 phosphorylation pathways were also disrupted in the Δ*Mocsn6* mutants. Furthermore, we found that MoCsn6 participates in the autophagy pathway by interacting with the autophagy core protein MoAtg6 and regulating its ubiquitination level. Deletion of *MoCSN6* resulted in rapid lipidation of MoAtg8 and degradation of the autophagic marker protein green fluorescent protein-tagged MoAtg8 under nutrient and starvation conditions, suggesting that MoCsn6 negatively regulates autophagic activity. Taken together, our results demonstrate that MoCsn6 plays a crucial role in regulating fungal development, pathogenicity, and autophagy in *M. oryzae*.

**IMPORTANCE** *Magnaporthe oryzae*, a filamentous fungus, is the cause of many cereal diseases. Autophagy is involved in fungal development and pathogenicity. The COP9 signalosome (CSN) has been extensively studied in ubiquitin pathways, but its regulation of autophagy has rarely been reported in plant-pathogenic fungi. Investigations on the relationship between CSN and autophagy will deepen our understanding of the pathogenic mechanism of *M. oryzae* and provide new insights into the development of new drug targets to control fungal diseases. In this study, the important function of Csn6 in the autophagy regulation pathway and its impact on the pathogenicity of *M. oryzae* were determined. We showed that Csn6 manages autophagy by interacting with the autophagy core protein Atg6 and regulating its ubiquitination level. Furthermore, future investigations that explore the function of CSN will deepen our understanding of autophagy mechanisms in rice blast fungus.

**KEYWORDS** COP9 signalosome, MoCsn6, autophagy, development, rice blast fungus

The COP9 (constitutive morphogenesis number 9) signalosome (CSN) is a highly conserved protein complex in eukaryotes that is indispensable for the development of animals and plants (1, 2). CSN was identified in 1996 as a negative regulator of light

**Ad Hoc Peer Reviewer** ⓘ Tong-Bao Liu, Southwest University

Address correspondence to Xiao-Hong Liu, xhliu@zju.edu.cn.

The authors declare no conflict of interest.

morphogenesis in *Arabidopsis thaliana* (3, 4). It consists of eight subunits (CSN1 to CSN8), each of which contains a characteristic domain PCI (proteasome lid-CSN-initiation factor 3) domain or an MPN (Mpr1 and Pad1 N-terminal) domain (5). More recently, CSN has been identified in various eukaryotes and shown to have complex biological functions, such as autophagy, circadian rhythm, signal transduction, T-cell development, cell cycle, embryonic development, and checkpoint control (6, 7). In addition to functioning as a complex, each subunit of CSN may have unique functions that contribute to its versatility (8–11). Maintaining protein homeostasis by regulating protein expression and degradation is essential for all aspects of cell development and proliferation. The autophagy-lysosomal system and the ubiquitin-proteasome system (UPS) are two major pathways for protein clearance in eukaryotes (12, 13). CSN, which is involved in multiple cellular and developmental processes, is a key regulator of the UPS. The most-studied function of CSN is to control the activity of cullin-RING E3 ubiquitin ligases (CRLs) (14, 15). CRL is the largest family of ubiquitin ligases and is responsible for approximately 20% of ubiquitin-dependent cellular protein degradation (16). The activation of CRL depends on the modification of cullins by the ubiquitin-like protein Nedd8 (neddylation). CSN acts as an isopeptidase and negatively regulates CRL activity by removing Nedd8 from cullins (deneddylation) (17). The recently identified ninth subunit of CSN, CSN9, contributes to the steric regulation of CRLs by reducing the affinity of CSN-CRL interactions, but it is not necessary for the assembly and catalytic activity of CSN (18). CSN9 stoichiometrically complexes with CSN1 to -8 to form a nine-membered noncanonical CSN complex (also known as CSN9-bound CSN) (19).

Autophagy, a highly conserved cellular pathway from yeast to mammalian eukaryotes, involves the delivery of organelles and long-lived proteins to vacuoles and lysosomes for degradation and recycling (20, 21). In recent years, increasing evidence has shown that moderate autophagy is crucial for fungal growth, development, and virulence (22, 23). Autophagy includes three essential steps: encirclement, delivery, and degradation, which require the synergy of core autophagy proteins (24). These core proteins constitute five systems necessary for normal autophagy, namely, the Atg13-Atg17-Atg1 initiation system, the Atg9-Atg2-Atg18 cycle system, the Atg12-Atg5-Atg16 ubiquitin-like system, the Atg8-PE ubiquitin-like system, and the phosphoinositide 3-kinase (PI3K) complex. Atg6 (homologous to Beclin1 and Vps30) is a subunit of the PI3K complex and plays an important role in autophagy and protein sorting (25). As the core protein of autophagy initiation, Atg6 acts as a scaffold during autophagy initiation and nucleation (26). Current findings suggest that Beclin1 mediates autophagosome formation by localizing various autophagy proteins to autophagy precursor structures and multiprotein interactions during autophagy initiation (27, 28). Beclin1 is regulated by many factors at the gene and protein levels, which may subsequently alter Beclin1-induced autophagy (26). Although the regulatory mechanism of Atg6 and its homolog has been revealed, its function in the pathogenesis of filamentous pathogenic fungi has not yet been explored.

*Magnaporthe oryzae*, the agent that causes rice blast disease, is considered the most destructive pathogenic fungus in the world (29, 30). Rice, wheat, and other gramineous plants are hosts of rice blast fungus. Because of its developmental characteristics and infection mechanism, *M. oryzae* has been used as a model organism to study the interaction between pathogenic fungi and host plants. Three-cell conidia, a powerful weapon of *M. oryzae*, grow on typical sympodial conidiophores and germinate to produce germ tubes under suitable environmental conditions (31). After that, the tip of the germ tube swells to form a dome-like infectious structure called an appressorium. The melanized appressorium produces a pressure of 8.0 MPa to directly destroy the plant cuticle and form infectious hyphae (32). The hyphae spread into adjacent plant cells and grow rapidly, resulting in typical water stains or yellowish-brown spots (33).

Although the importance of Csn6 as a CSN subunit has been demonstrated in different species, its function in filamentous fungi has not been investigated to our knowledge. In this study, we identified the homolog of Csn6 in *M. oryzae* and named it MoCsn6. Our results suggest that MoCsn6 negatively regulates autophagy by interacting with the

autophagy core protein MoAtg6 and regulating its ubiquitination level. In addition, phenotypic analysis and Western blot assays confirmed that MoCsn6 is essential for vegetative growth, conidial production, the Osm1 and Pmk1 phosphorylation pathways, and the virulence of *M. oryzae*. In conclusion, we demonstrate the biological function of MoCsn6, a subunit of the COP9 signalosome, for the first time and show that it plays an essential role in autophagy and ubiquitination.

## RESULTS

**Identification of the COP9 signalosome in *M. oryzae* and interaction networks of the COP9 signalosome.** Previous studies have shown that the autophagy process relies on core ATG (autophagy-related) proteins, which are involved in the initiation, extension, and formation of autophagosomes (22, 24). To identify novel proteins associated with autophagy, we screened rice blast fungus cDNA libraries using the autophagy core protein MoAtg6 as bait. A total of 109 proteins were identified by two screenings. To clarify the functions of these proteins, 76 corresponding genes were deleted using high-throughput gene knockout strategies (see Table S1 in the supplemental material). We found that a hypothetical protein that is encoded by the gene MGG_01432 and involved in the autophagy pathway appeared in the two screening results. The amino acid sequences were used for alignment in the NCBI databases to identify the hypothetical protein. The results showed that this protein is subunit 6 of CSN and contains a conserved MPN domain, which is highly conserved in eukaryotes (Fig. 1A). The "coverage" is a parameter of the sequence alignment result, i.e., the coverage of the submitted sequence relative to the target sequence. In our study, this protein had 41.19% amino acid homology and 93% coverage compared with Csn6 of *Neurospora crassa* (Fig. 1B); thus, we named the protein MoCsn6.

In contrast to CSN in plants and animals, CSN is unnecessary for fungal survival (34). Therefore, fungi are suitable models for exploring the composition, cellular function, and activity of CSN, but CSN has not been studied in the plant-pathogenic fungus *M. oryzae*. We identified seven conserved CSN subunits with mammalian homology in *M. oryzae* by using the NCBI and UniProt databases. Among them, Csn6 and Csn5 contained a conserved MPN domain, while Csn1 to -4 and Csn7a contained a conserved PIC domain (5, 8) (Fig. 2A). However, the mammalian homolog for Csn8 was not found in *M. oryzae*. Next, we detected interactions between CSN subunits based on the yeast two-hybrid system. Due to the self-activation of MoCsn2-AD, MoCsn2-BD, MoCsn3-BD, and MoCsn5-AD, we did not detect protein interactions based on the above four vectors. As shown in Fig. 2B, our experimental results showed that MoCsn1 could interact with MoCsn3, MoCsn4, MoCsn5, and MoCsn6 and that MoCsn4 could interact with MoCsn5, MoCsn6, and MoCsn7a. There was also a weak interaction between MoCsn3 and MoCsn4, MoCsn5, MoCsn6, and MoCsn7a. Interactions between MoCsn5 and MoCsn6, MoCsn5 and MoCsn7a, and MoCsn6 and MoCsn7a were observed. However, there was no interaction between MoCsn1 and MoCsn7a (Fig. 2B). Furthermore, the results of the coimmunoprecipitation (co-IP) experiments showed that MoCsn6 could interact with MoCsn1, MoCsn3, MoCsn4, MoCsn5, and MoCsn7a *in vivo* to help CSN complete its biological functions (Fig. S1).

**MoCsn6 is mainly localized in the nucleus and cytoplasm of *M. oryzae*.** To explore the subcellular localization of the MoCsn6 protein in *M. oryzae*, we constructed a MoCsn6-green fluorescent protein (GFP) vector and transformed it into the Δ*Mocsn6* mutant. Under a fluorescence microscope, the signals of MoCsn6-GFP appeared in the nucleus-like structure and cytoplasm. To further elucidate the localization of MoCsn6 at different developmental stages of *M. oryzae*, an MoH$_2$B-mCherry (nuclear location marker) fusion protein was coexpressed with MoCsn6-GFP in the Δ*Mocsn6* mutant. As shown in Fig. 3A and B, each cell of the conidia, appressorium, and hyphae contained a dominant MoCsn6-GFP fluorescent spot that overlapped with the MoH$_2$B-mCherry signal. In addition, the fluorescent signal of MoCsn6-GFP was also dispersed in the cytoplasm (Fig. 3). Therefore, we concluded that MoCsn6 is predominantly localized in the nucleus and cytoplasm of *M. oryzae*.

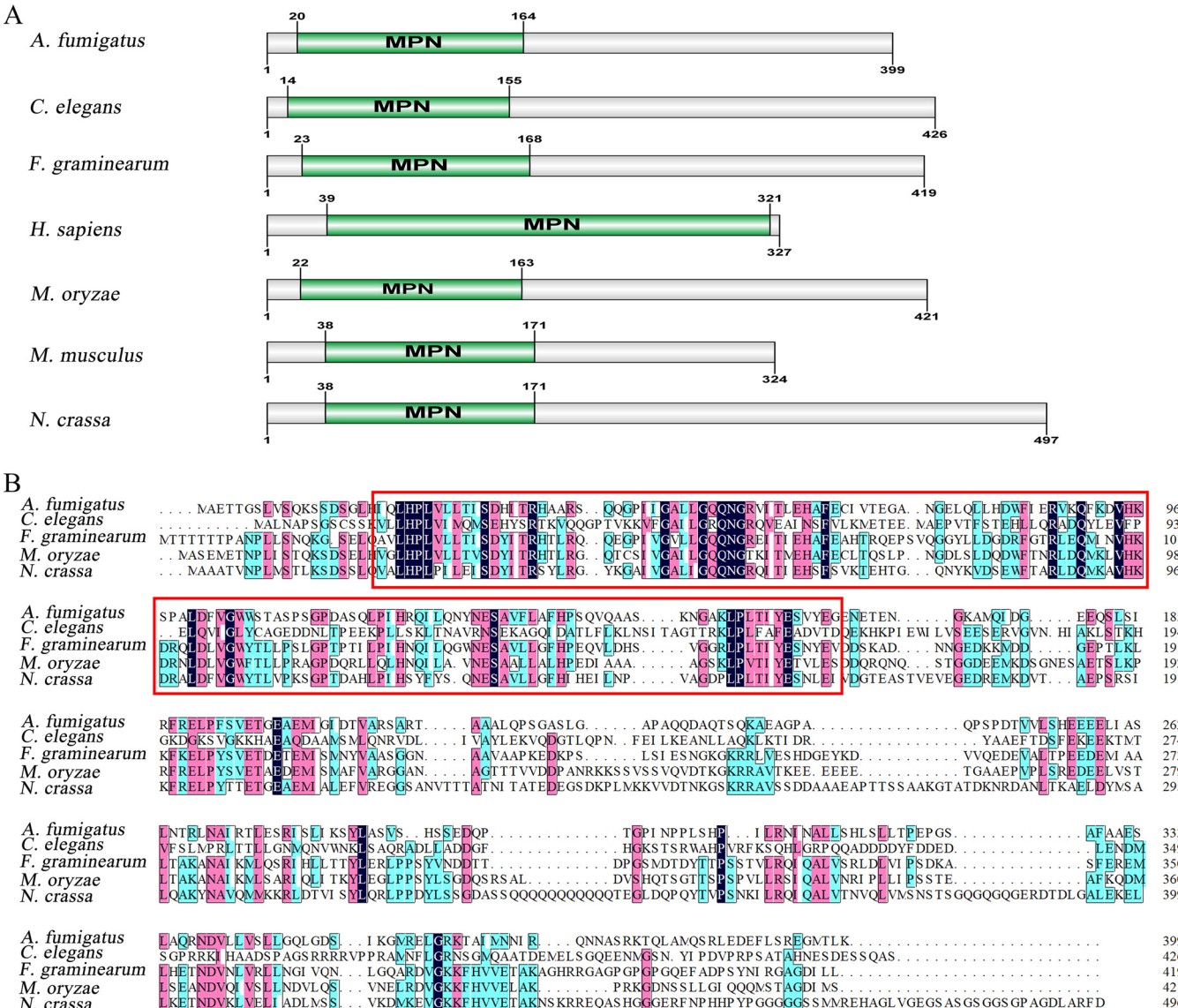

**FIG 1** Comparison of Csn6 amino acid sequences and MPN domains in different eukaryotes. (A) MPN domain diagram of Csn6 in *M. oryzae*, *F. graminearum*, *A. fumigatus*, *C. elegans*, *Mus musculus*, *N. crassa*, and *H. sapiens*. (B) Sequence alignment of Csn6 amino acid sequences in different eukaryotes was performed using DNAMAN v.8 software. All of these Csn6 proteins contained a conserved MPN domain (red framed part).

**MoCsn6 is required for vegetative growth and conidiation.** To assess the physiological function of MoCsn6 in *M. oryzae*, we deleted the *MoCSN6* gene in the wild-type strain 70-15 via the target gene replacement method (Fig. S2). Four deletion mutants with similar phenotypes were obtained. We selected one of the mutants, named Δ*Mocsn6*, and constructed the complemented strain *Mocsn6c* (Δ*Mocsn6* supplemented with MoCsn6-GFP) for further analysis. The differences in colony morphology, vegetative growth, and conidiation among 70-15, Δ*Mocsn6*, and *Mocsn6c* were characterized by phenotypic analysis.

First, we examined the colony morphology of the wild-type strain 70-15, Δ*Mocsn6* mutant, and complemented strain *Mocsn6c*. The vegetative growth of Δ*Mocsn6* was slower than that of 70-15 and *Mocsn6c* (Fig. 4A and C). Compared with 70-15, the colony diameter of the Δ*Mocsn6* mutant was 40.4% lower on complete medium (CM) and 22.3% lower on minimal medium (MM), and the aerial hyphae of the mutant were sparser than those of 70-15 (Fig. 4B, C, and D). As shown in Fig. 4A and B, the bottom of the Δ*Mocsn6* mutant colony was dark brown, which differed from the light yellow color of 70-15 and *Mocsn6c*. Moreover,

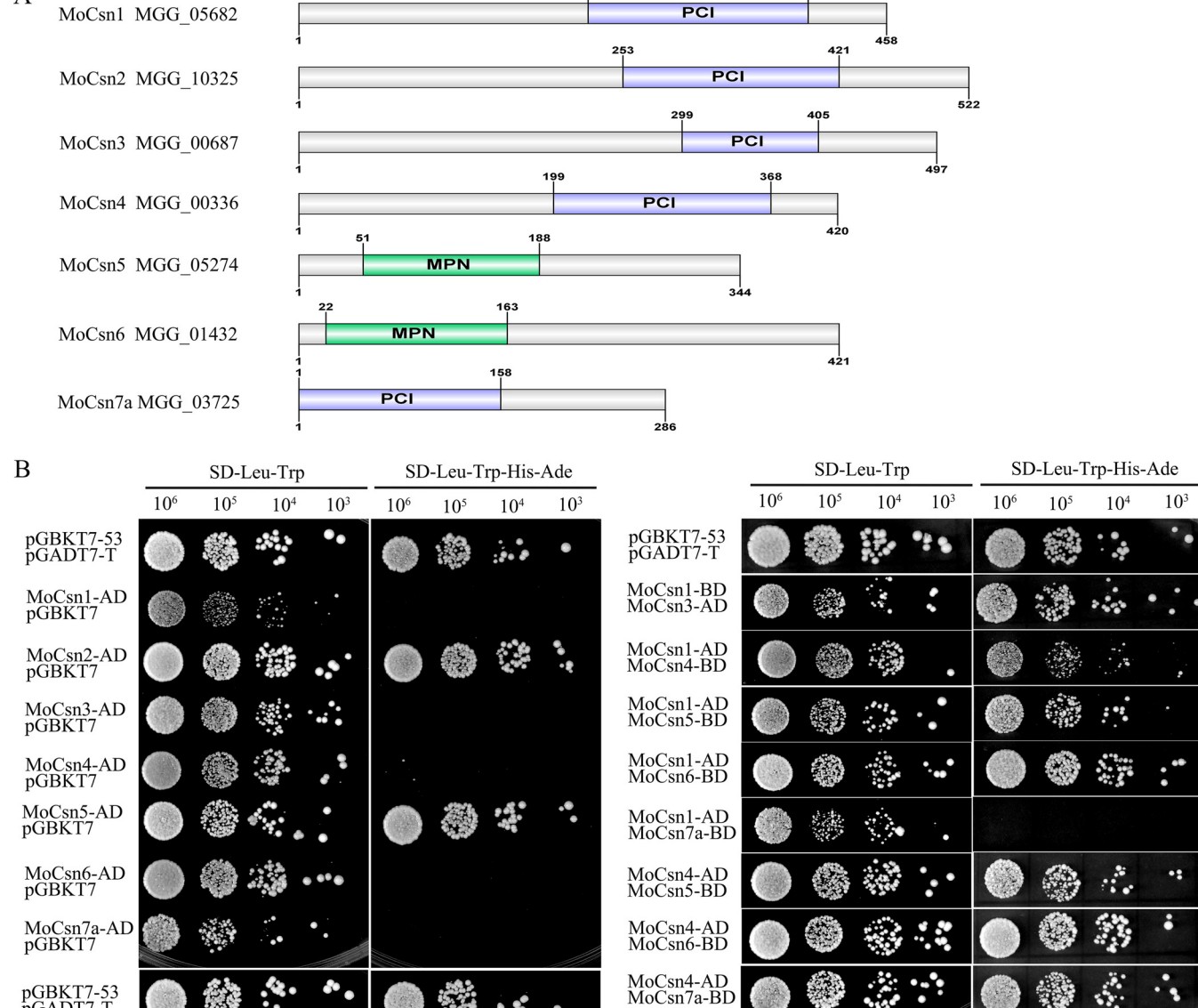

**FIG 2** Conserved homologs of CSN in *M. oryzae* and interactions of the CSN subunits based on the yeast two-hybrid experiments. (A) Sequences of CSN subunits were obtained from the NCBI databases. Conserved domain analysis of identified homologs was performed on the websites https://www.uniprot .org/. The picture was drawn with IBS v1.0 software. (B) Interactions between CSN subunits based on the yeast two-hybrid experiments. The experiment was repeated independently for three times, and the representative pictures were displayed.

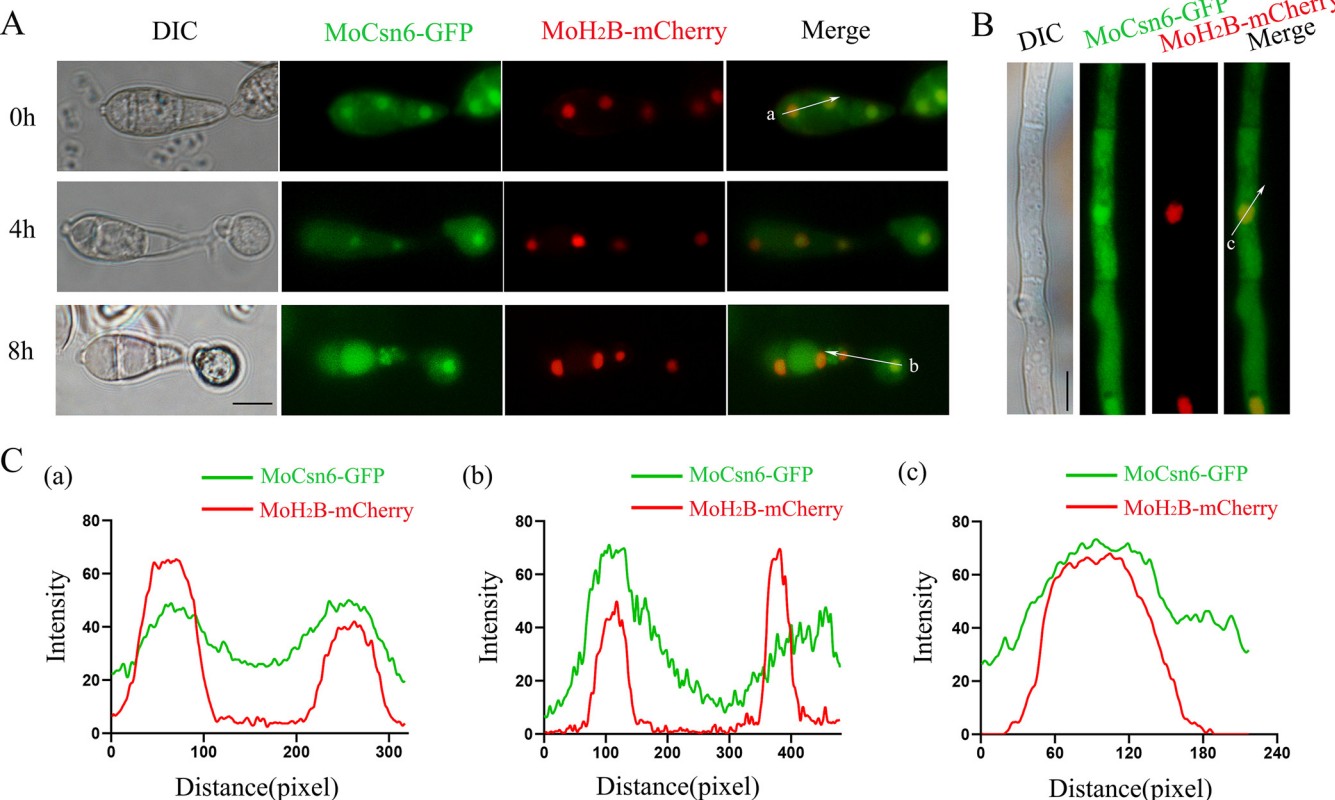

**FIG 3** MoCsn6 is localized to the nucleus and cytoplasm. (A) The localization of MoCsn6 in conidia and appressoria. MoH$_2$B-mCherry and MoCsn6-GFP were coexpressed in the Δ*Mocsn6* mutant. Conidia were washed from colonies that were grown in CM for 10 days. The conidia suspension (5 × 10$^4$ mL$^{-1}$) was placed on hydrophobic plastic film and incubated in a humid chamber at 25°C. Fluorescence in spores and appressoria was observed under a fluorescence microscope at 0, 4, and 8 hpi. Bar, 10 $\mu$m. (B) The localization of MoCsn6 in hyphae. Bar, 10 $\mu$m. (C) ImageJ software was used to analyze the fluorescence densities of MoH$_2$B-mCherry and MoCsn6-GFP.

the colonies of 70-15 and *Mocsn6c* were round with smooth edges, while Δ*Mocsn6* was irregular with prominent aerial mycelia at the edges (Fig. 4A). Next, we examined the conidiophores of 70-15, Δ*Mocsn6*, and *Mocsn6c*. The wild-type 70-15 and *Mocsn6c* strains had typical sympodial conidiophores bearing multiple spores on each conidiophore, while the Δ*Mocsn6* mutant lacked fascicular conidiophores and showed a single-branching pattern (Fig. 4D). Quantitative analysis showed that conidiation in the Δ*Mocsn6* mutant strain was dramatically diminished by 40.73-fold. Strain 70-15 produced (4.44 ± 0.21) × 10$^4$ conidia/cm$^2$ (mean ± standard deviation), while the Δ*Mocsn6* mutant produced (0.11 ± 0.02) × 10$^4$ conidia/cm$^2$ (Fig. 4D and E). In conclusion, the colony morphology, vegetative growth, and conidiation of Δ*Mocsn6* were severely impaired, implying that MoCsn6 participates in the development of *M. oryzae*.

**Conidial germination and appressorium formation are impaired in the Δ*Mocsn6* mutant.** Since normal appressorium formation is a prerequisite for rice blast pathogenicity, we examined conidia germination and appressorium formation in wild-type 70-15, the Δ*Mocsn6* mutant, and *Mocsn6c*. The deletion of *MoCSN6* did not affect conidia morphology or size (Fig. 5A). The conidia of Δ*Mocsn6* were typical three-celled conidia, similar to those of 70-15 and *Mocsn6c*. However, the appressorium formation rate was >90% in 70-15 and *Mocsn6c* at 8 h postinoculation (hpi) and 24 hpi, while it was only 4.8% in the Δ*Mocsn6* mutant at 8 hpi and 7.4% at 24 hpi; the conidial germination rate of Δ*Mocsn6* was only 22.8% at 8 hpi and 30.1% at 24 hpi (Fig. 5A, B and C). These results showed that conidia germination and appressorium formation are impaired in the Δ*Mocsn6* mutant. Because the mitogen-activated protein kinase (MAPK) signaling pathway has been shown to be critical for appressorium formation and the virulence of *M. oryzae* (35, 36), we analyzed Pmk1 activation in the mycelia of 70-15 and Δ*Mocsn6* by Western blot analysis. As shown in Fig. 5D, the

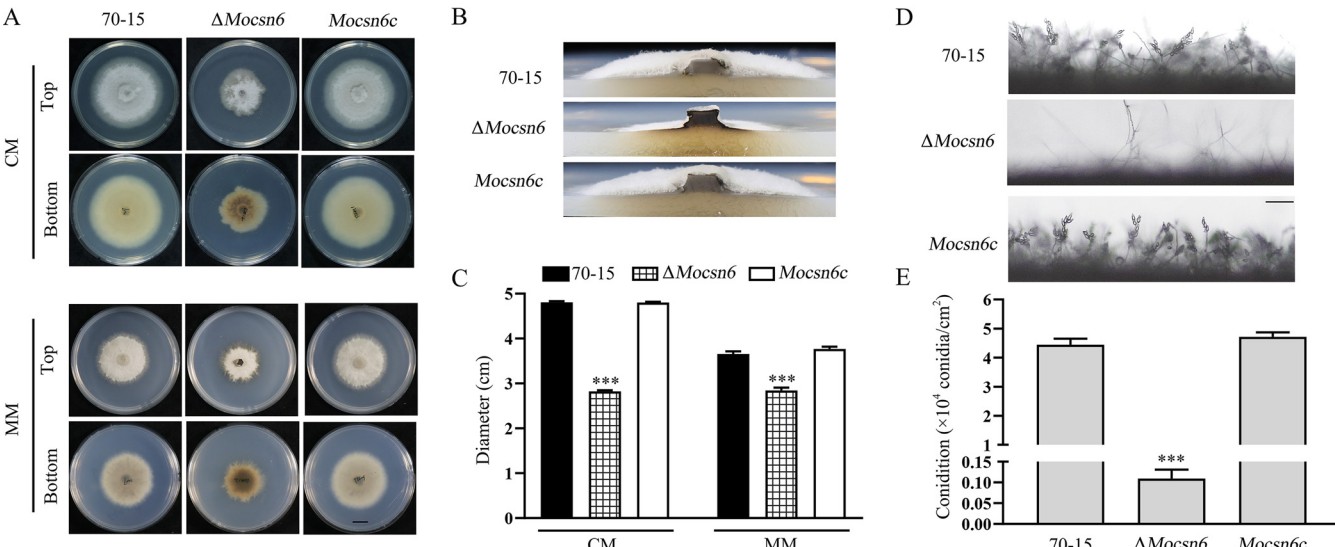

**FIG 4** MoCsn6 is required for vegetative growth and conidiation. The experiments were repeated independently three times, and representative pictures are displayed. (A and B) Colony morphology and color of the 70-15 strain, ΔMocsn6 mutant, and the complementation strain Mocsn6c in CM and MM at 25°C for 9 days. Bar, 1 cm. (C) Colony diameters of 70-15, ΔMocsn6, and Mocsn6c. The data were analyzed with GraphPad Prism 8.0 software. ***, $P < 0.001$. (D) Conidiophores and conidia of 70-15, ΔMocsn6, and Mocsn6c. Bar, 100 $\mu$m. (E) Statistical analysis of conidia production. The data were analyzed with GraphPad Prism 8.0 software. ***, $P < 0.001$.

phosphorylation level of Pmk1 was higher in ΔMocsn6 compared with 70-15. The results showed that the abnormal phosphorylation of Pmk1 was one of the reasons why ΔMocsn6 failed to form appressorium.

**MoCsn6 is essential for virulence in *M. oryzae*.** Rice blast fungus is a model organism used to study the interaction between host plants and pathogens, and appressoria are typical infection structures of *M. oryzae* (30, 37). To further investigate the role

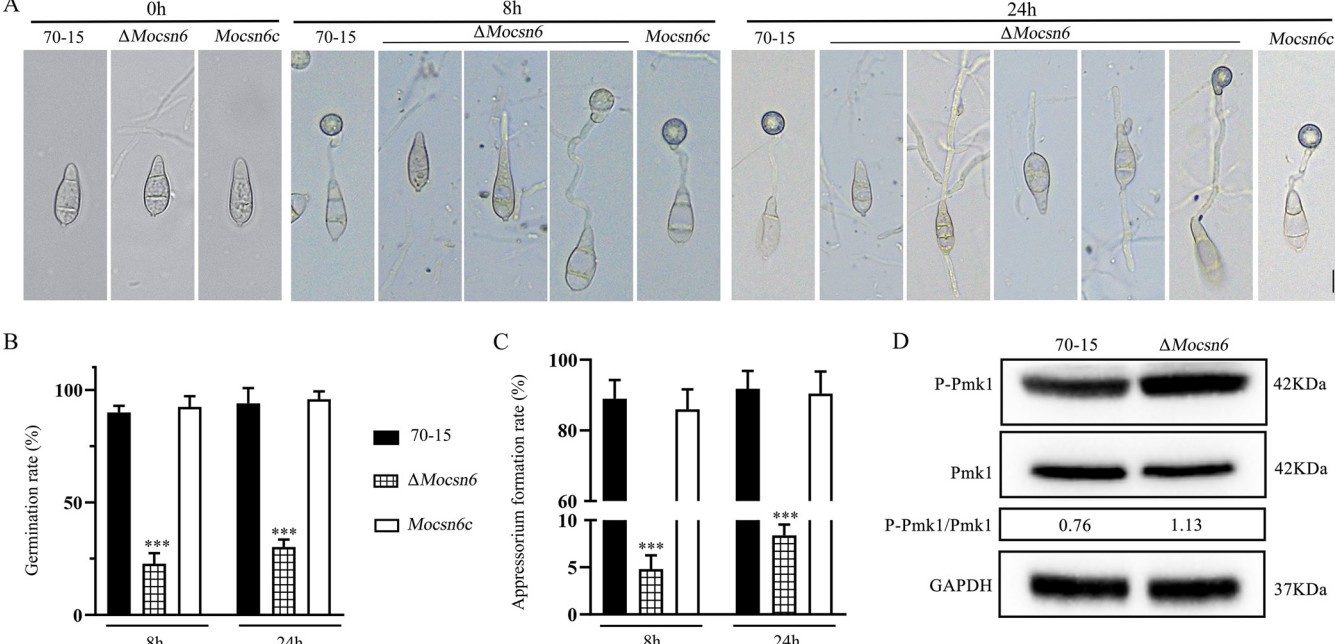

**FIG 5** Conidial germination and appressorium formation were impaired in the ΔMocsn6 mutant. (A) Conidial morphology and appressorium morphology of 70-15, ΔMocsn6, and Mocsn6c. Appressorium formation assays were carried out on hydrophobic surfaces. Bar, 10 $\mu$m. (B) Germ tube germination rates of 70-15, ΔMocsn6, and Mocsn6c were statistically analyzed with GraphPad Prism 8.0 software. ***, $P < 0.001$. (C) Appressorium formation rates of 70-15, ΔMocsn6, and Mocsn6c were statistically analyzed with GraphPad Prism 8.0 software. ***, $P < 0.001$. (D) The phosphorylation level of Pmk1 was detected by Pmk1 and phospho-Pmk1 antibodies. The protein content of GAPDH was used as a control.

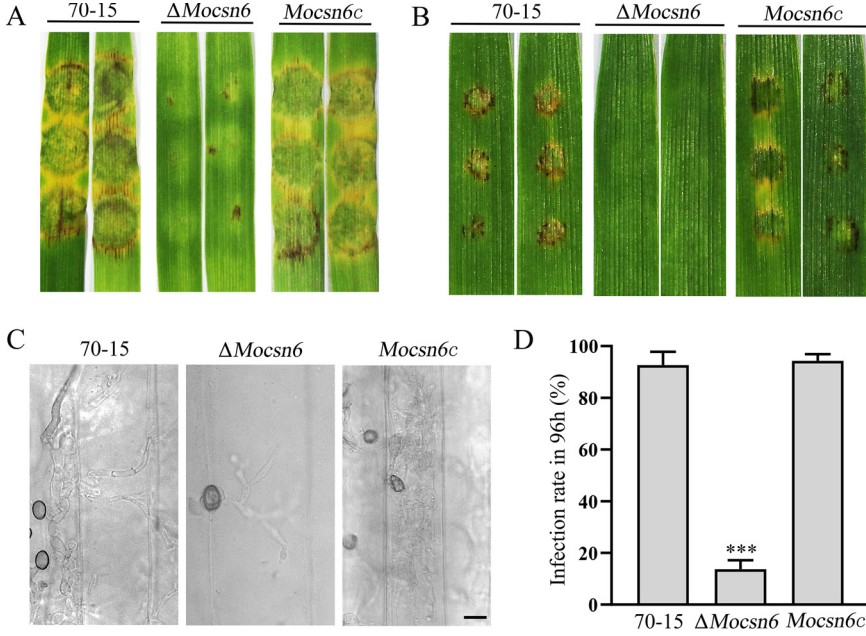

**FIG 6** MoCsn6 is essential for virulence in *M. oryzae*. (A) Disease symptoms on isolated leaves of barley inoculated with mycelial plugs from 70-15, Δ*Mocsn6*, and *Mocsn6c*. Photographs were taken 4 days after inoculation. (B) Disease symptoms on isolated leaves of barley inoculated with conidial suspension ($5 \times 10^4$ mL$^{-1}$) from 70-15, Δ*Mocsn6*, and *Mocsn6c*. Photographs were taken 4 days after inoculation. (C) Penetration assays were carried out on isolated leaves. After 96 h, the leaves were decolorized and observed under an optical microscope. Bar, 15 μm. (D) Penetration rate of appressoria of 70-15, Δ*Mocsn6*, and *Mocsn6c* at 96 hpi. ***, $P < 0.001$.

of MoCsn6 in the pathogenicity of *M. oryzae*, systematic pathogenicity analysis assays were carried out.

First, mycelial plugs of 70-15, Δ*Mocsn6*, and *Mocsn6c* were inoculated on isolated leaves of barley grown for 8 days. After 4 days, 70-15 and *Mocsn6c* caused typical yellow-brown lesions with rotting plant tissue in the center, while the Δ*Mocsn6* mutant produced only a few dark spots without serious lesions (Fig. 6A). In addition, we inoculated the wounded barley and rice leaves with mycelial plugs of three strains (70-15, Δ*Mocsn6* mutant, and *Mocsn6c*) for 3 days. Strain 70-15 and *Mocsn6c* caused severe lesions, while the Δ*Mocsn6* mutant resulted in small disease lesions (Fig. S3A and B). As mentioned above, the sporulation quantity of Δ*Mocsn6* was significantly reduced; we thus speculated that the reduced pathogenicity was due to the low level of conidiation.

Next, we investigated the pathogenicity of conidia. The conidial suspensions of 70-15, Δ*Mocsn6*, and *Mocsn6c* were adjusted to the same concentration ($5 \times 10^4$ mL$^{-1}$) and inoculated on isolated leaves of barley. Interestingly, after 4 days, the leaves inoculated with spore suspensions of 70-15 and *Mocsn6c* showed severe damage, while the leaves inoculated with spore suspensions of Δ*Mocsn6* showed no obvious lesions (Fig. 6B). Moreover, a penetration assay on plant surface cells was performed. After 96 h, the appressorium infection rate of the Δ*Mocsn6* mutant was only 12.3%, which was significantly lower than the 93.4% rate of 70-15 (Fig. 6D). The infective hyphae of 70-15 and *Mocsn6c* expanded into multiple plant cells, while the infective hyphae of Δ*Mocsn6* were limited to a single plant cell (Fig. 6C). These results indicated that MoCsn6 is required for the pathogenicity of *M. oryzae*.

**MoCsn6 plays a role in responses to hyperosmotic stresses.** To elucidate whether MoCsn6 is involved in fungal tolerance to exogenous stress, 70-15, Δ*Mocsn6*, and *Mocsn6c* were inoculated in solid CM supplemented with hypertonic stress drugs to analyze their growth defects. Compared with 70-15, Δ*Mocsn6* showed higher sensitivity in medium supplemented with 0.4 M KCl, 0.6 M NaCl, or 0.8 M sorbitol, and the growth rate decreased by 26.5%, 24.4%, and 23.1%, respectively (Fig. 7A and B). These

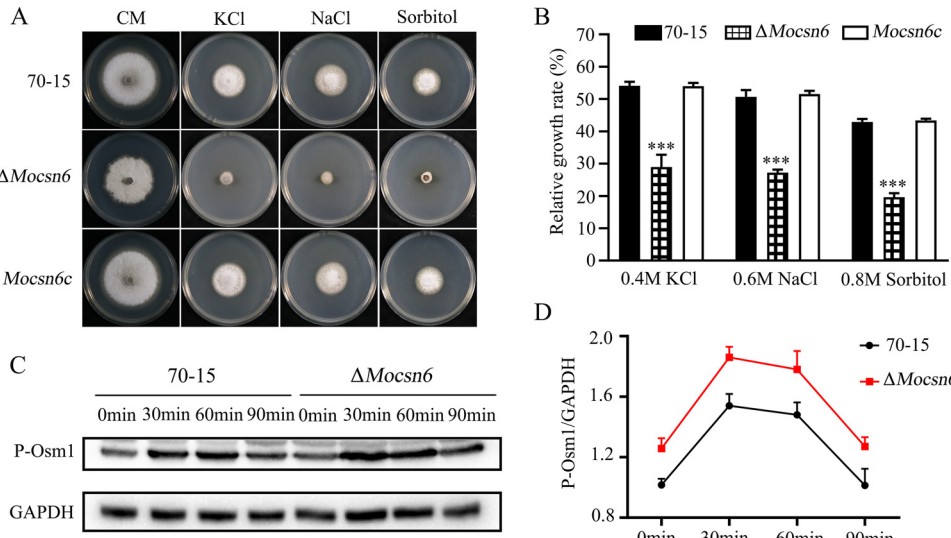

**FIG 7** MoCsn6 plays a role in responses to hyperosmotic stresses. (A) Colony morphology of 70-15, Δ*Mocsn6*, and *Mocsn6c* on CM with 0.4 M KCl, 0.6 M NaCl, or 0.8 M sorbitol. Pictures were taken at 8 days. (B) Relative growth rates of 70-15, Δ*Mocsn6*, and *Mocsn6c* under different hyperosmotic stresses. The data were analyzed with GraphPad Prism 8.0 software. ***, $P < 0.001$. (C) The phosphorylation level of Osm1 in 70-15 and Δ*Mocsn6* was detected by the anti-phospho-Osm1 antibody. The protein content of GAPDH was used as a control. (D) GraphPad Prism 8.0 software was used to analyze the phosphorylation level of Osm1.

data indicate that MoCsn6 is necessary for adapting to hyperosmotic stress in the rice blast fungus.

In *M. oryzae*, the MAPK Osm1-mediated signaling pathway is responsible for regulating the hypertonic stress response (33). To verify whether MoCsn6 regulates this pathway by affecting the phosphorylation of Osm1, we further examined the Osm1 MAPK signaling pathway. When strain 70-15 and the Δ*Mocsn6* mutant were grown in CM for 40 h without the addition of 0.6 M NaCl, the phosphorylation level of Osm1 was very low. As shown in Fig. 7C and D, after treatment with 0.6 M NaCl, the Osm1 phosphorylation level in 70-15 and Δ*Mocsn6* increased before 30 min and then decreased later. In contrast, the phosphorylation levels of Δ*Mocsn6* were always higher than 70-15 before 90 min. The above results suggest that MoCsn6 is involved in controlling the Osm1 phosphorylation level in response to hyperosmotic stress.

**MoCsn6 interacts with MoAtg6 *in vivo* and *in vitro*.** To further clarify the interaction between MoCsn6 and core ATG proteins, we performed yeast two-hybrid assays. The MoCsn6 cDNA fragment was fused into the pGADT7 vector, and the cDNA fragments of 22 ATG proteins were fused into the pGBKT7 vector. The results showed that yeast cotransformed with MoCsn6-AD and MoAtg6-BD was similar to the positive control and could still grow in SD/-Leu-Trp-His-Ade yeast-deficient medium (Fig. 8A), indicating that MoCsn6 interacts with MoAtg6 *in vitro*. The interaction of MoCsn6 and MoAtg6 was also confirmed by the co-IP assay *in vivo*. MoAtg6-GFP or GFP was separately transformed into the Δ*Mocsn6* mutant complemented with MoCsn6-Flag. The MoCsn6-Flag band was detected in the MoAtg6-GFP coimmunoprecipitation lysate, but it was not found in the GFP coimmunoprecipitation lysate (Fig. 8B). These results show that MoCsn6 interacts with MoAtg6 in *M. oryzae*.

**MoCsn6 negatively regulates autophagy.** Autophagy is necessary for sexual and asexual reproduction, vegetative growth, and complete virulence of plant-pathogenic fungi (24, 38). Research has shown that the storage of energy, formation of infected nails, and adaptation to stress environments depend on moderate autophagy (22, 23). When autophagy is induced, a reliable autophagy marker Atg8 is linked to phosphatidylethanolamine (PE) to form Atg8-PE (a process called Atg8 lipidation), which is anchored to the autophagosome membrane and transferred to vacuoles as the autophagosome degrades (21). Atg8-PE is an autophagy reporter that is reliably associated with completed autophagosomes. Changes in the amount of Atg8-PE are closely related to changes in the number of autophagosomes

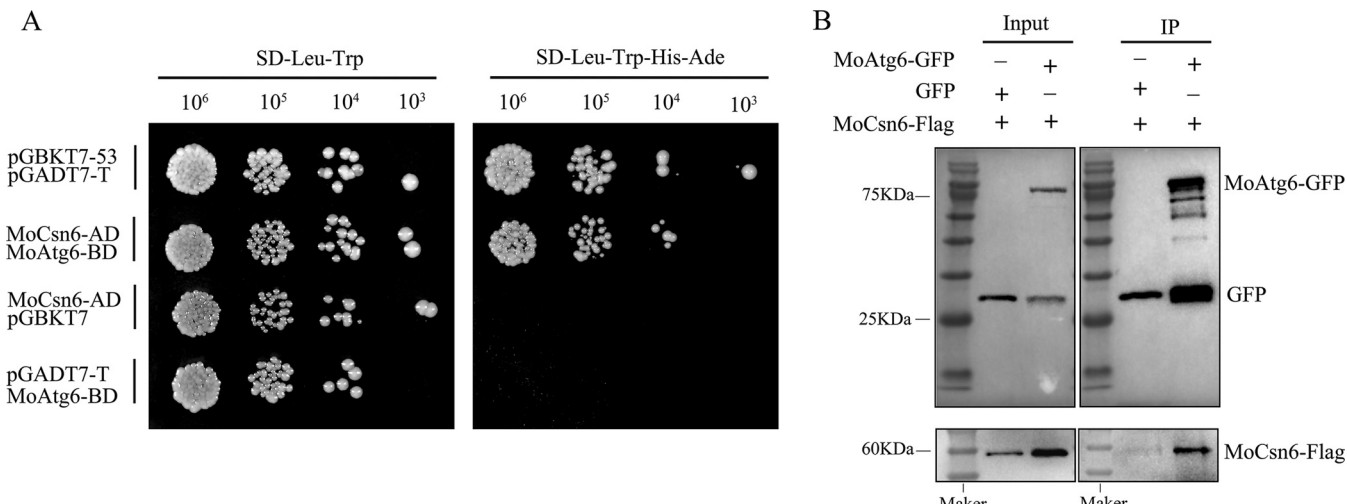

**FIG 8** MoCsn6 interacts with MoAtg6 *in vivo* and *in vitro*. (A) The interaction between MoCsn6 and MoAtg6 was detected by yeast two-hybrid assays. pGADT7-T and pGBKT7-53 were used as positive controls. The two pairs of negative controls were pGADT7-T and MoAtg6-BD and pGBKT7 and MoCsn6-AD. (B) The interaction between MoCsn6 and MoAtg6 was detected by co-IP. MoAtg6-GFP and MoCsn6-Flag were coexpressed in the Δ*Mocsn6* mutant. Flag antibody and GFP antibody were used to detect the expression of the GFP label in transformants by Western blotting.

present in cells (39). The final step in the autophagy process is the degradation of autophagosomes after fusing with lysosomes or vacuoles. Since MoCsn6 was discovered in rice blast cDNA libraries using core ATG proteins and interacts with MoAtg6, we further explored its role in the autophagy process of *M. oryzae*.

To further investigate the relationship between MoCsn6 and autophagy, we first analyzed the endogenous lipidation of MoAtg8 in 70-15 and the Δ*Mocsn6* mutant strain by Western blot assays. As shown in Fig. 9A and B, under nutrient conditions, the Atg8 bands were strong in 70-15 and Δ*Mocsn6*, and the Atg8-PE bands were weak; however, the content of Atg8-PE in 70-15 was lower than that in Δ*Mocsn6*. After induction in nutrient-limited medium without amino acids and ammonium sulfate (SD-N) for 3 and 6 h, the Atg8-PE content of 70-15 and Δ*Mocsn6* was significantly increased compared with that under nutrient conditions. Compared with 70-15, the levels of MoAtg8-PE and MoAtg8 were significantly increased in Δ*Mocsn6* (Fig. 9A and B). These results indicated that the conversion of MoAtg8 to MoAtg8-PE proceeded normally in Δ*Mocsn6*, but the total amount was increased. We also observed autophagosomes in the mycelia of strain 70-15 and the Δ*Mocsn6* mutant under nutrient conditions and starvation conditions by light microscopy. The results also showed that the number of autophagosomes in the Δ*Mocsn6* mutant was significantly greater than that in 70-15 (Fig. S4).

It is well known that autophagy is a dynamic process; thus, the accumulation of autophagosomes may be caused by increased autophagic flux or blocked autophagy. To clarify the cause of the increased content of MoAtg8 and MoAtg8-PE in Δ*Mocsn6*, we expressed the GFP-MoAtg8 fusion protein in 70-15 and Δ*Mocsn6* and analyzed the degradation process of GFP-MoAtg8 by fluorescence microscopy and Western blotting. As shown in Fig. 9E, autophagy did not occur in 70-15 under nutrient conditions, and the GFP-MoAtg8 fusion protein was distributed in the cytoplasm. After induction under starvation conditions for 3 h, the GFP-MoAtg8 fusion protein in 70-15 was transferred to the vacuole and degraded, but not completely; a few GFP-MoAtg8 puncta were observed in the cytoplasm (Fig. 9E). Compared with the wild-type strain, 70-15, few GFP-MoAtg8 puncta were observed in Δ*Mocsn6* under nutrient conditions, and fluorescence was evident in the vacuolar lumen. Under starvation conditions, the fluorescence of GFP-MoAtg8 was almost completely located in the vacuole, and there were almost no visible fluorescent spots in the Δ*Mocsn6* mutant (Fig. 9E).

Next, a Western blot assay was used to detect the fused GFP-MoAtg8 bands and free GFP bands, which corresponded to the fluorescence of unentered and entered

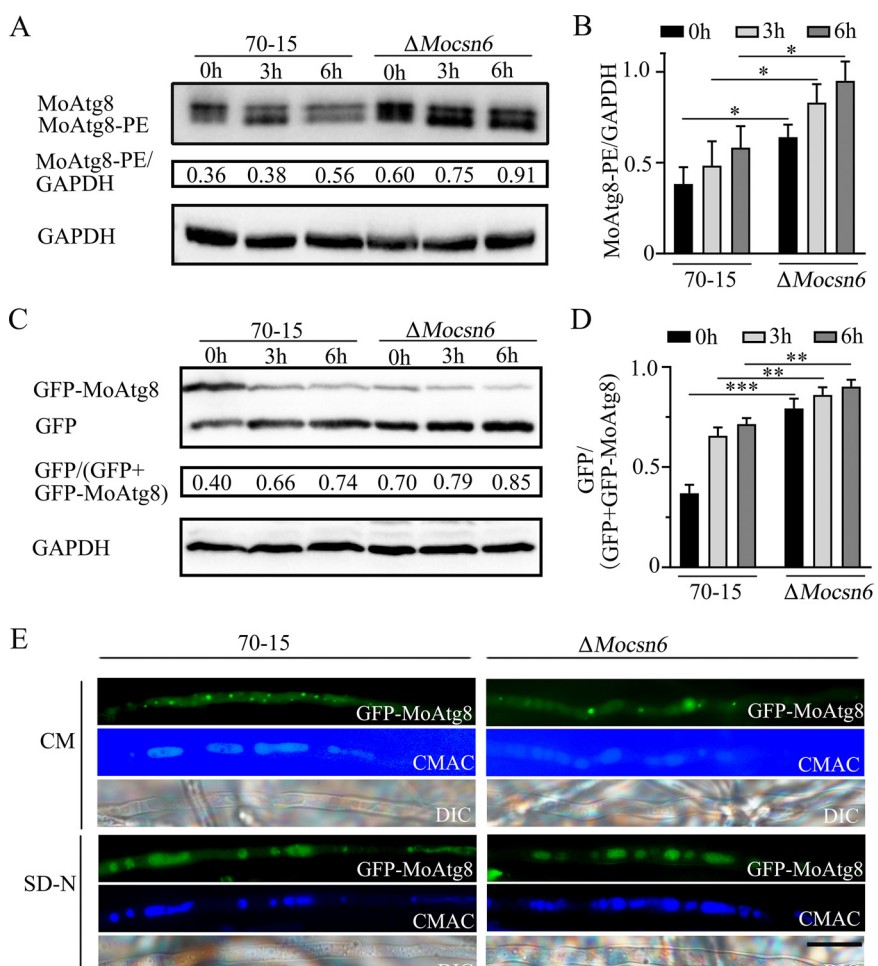

**FIG 9** MoCsn6 negatively regulates autophagy. (A) MoAtg8/MoAtg8-PE turnover was detected in 70-15 and ΔMocsn6 mutants under nitrogen starvation for 0, 3, and 6 h by Western blotting. The protein content of GAPDH was used as a control. (B) ImageJ and GraphPad Prism 8.0 software were used to analyze the MoAtg8-PE GAPDH in 70-15 and ΔMocsn6. *, $P < 0.05$. (C) GFP-MoAtg8 proteolysis was detected in 70-15 and ΔMocsn6 mutants under nitrogen starvation for 0, 3, and 6 h by Western blotting. The degradation rate was expressed as [GFP/(GFP+GFP-MoAtg8)]. The protein content of GAPDH was used as a control. (D) ImageJ and GraphPad Prism 8.0 software were used to analyze the rate GFP/(GFP+GFP-MoAtg8) in 70-15 and ΔMocsn6. ***, $P < 0.001$; **, $P < 0.01$. (E) Localization of GFP-MoAtg8 in the 70-15 strain and ΔMocsn6 mutants. Strains were grown in liquid CM at 150 rpm for 36 h and then shifted to liquid SD-N medium at 180 rpm for 3 h. The mycelia were stained with CMAC and observed under a fluorescence microscope. Bar, 10 μm.

vacuoles. The degradation of GFP-MoAtg8 indicated the rate of autophagosome fusion with the vacuole. The ratio of [(GFP)/(GFP+GFP-MoAtg8)] was used to quantify the autophagic flux. When hyphae were cultured in CM for 40 h, the GFP-MoAtg8 band was strong in 70-15 and the free GFP band was weak; the free GFP band was strong in the ΔMocsn6 mutants and the GFP-MoAtg8 band was weak (Fig. 9C and D). When the hyphae were transferred to SD-N medium for 3 and 6 h, the GFP-MoAtg8 bands in 70-15 became weak and the free GFP band became strong; the fusion bands almost disappeared in the ΔMocsn6 mutant (Fig. 9C and D). Western blot assays showed that the degradation rate of GFP-MoAtg8 in the ΔMocsn6 mutant was significantly faster than that of 70-15 under nutrient and starvation conditions (Fig. 9C and D). In general, the deletion of *MoCSN6* results in increased autophagic flux and faster autophagy. MoCsn6 negatively regulates autophagy in *M. oryzae*.

**MoCsn6-mediated ubiquitination pathways.** Due to the special role of CSN in ubiquitination-deubiquitination and the confirmed interaction of MoCsn6 with MoAtg6, we analyzed the role of MoCsn6 in overall ubiquitination and MoAtg6 ubiquitination using hyphal

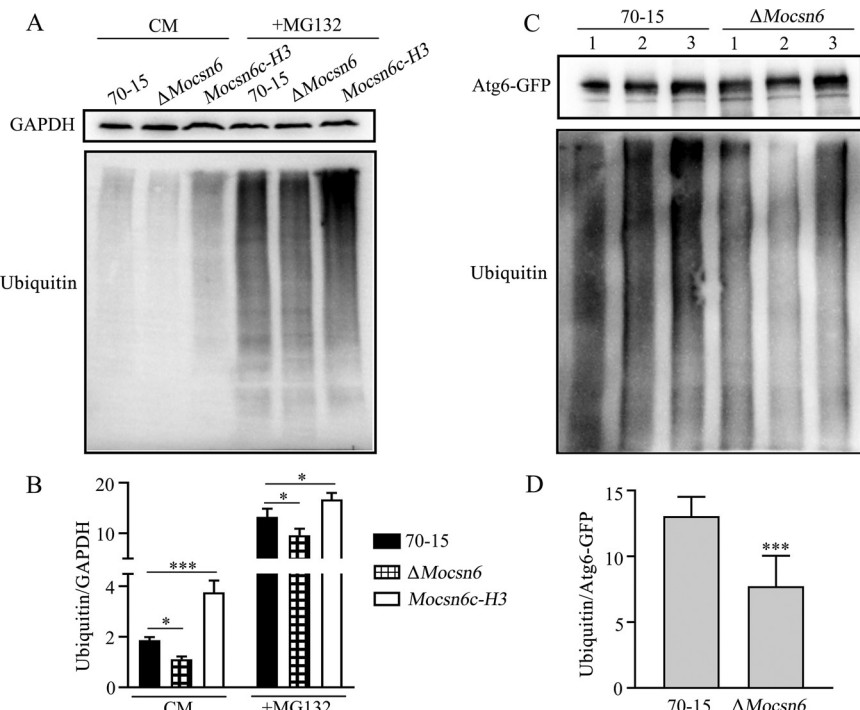

**FIG 10** MoCsn6 mediates ubiquitination pathways. (A) Protein ubiquitination levels were reduced in Δ*Mocsn6* mutants. Hyphae were cultured in CM for 36 h and then transferred to CM supplemented with MG132 (a proteasome inhibitor, a tool for studying cellular degradation of the ubiquitin-proteasome pathway) for 0 or 4 h. (B) Protein ubiquitination levels of 70-15, Δ*Mocsn6* mutant, and *Mocsn6c-H3*. ***, $P < 0.001$; *, $P < 0.05$. (C) Deletion of *MoCSN6* inhibits the ubiquitination of MoAtg6. GFP-Atg6 was expressed in 70-15 and Δ*Mocsn6* cells, pulldown analysis was performed with GFP beads, and immunoblot analysis was performed with an anti-polyubiquitin antibody. (D) The ubiquitination level of MoAtg6 in 70-15 and Δ*Mocsn6* mutants. ***, $P < 0.001$.

protein extracts from wild-type 70-15, Δ*Mocsn6* mutant, and compensatory strain *Mocsn6c-H3* (Δ*Mocsn6* supplemented with strong promoter H3). The strain with strong promoter H3 was used to detect the effect of high MoCsn6 expression on the overall ubiquitination level. Although similar in glyceraldehyde 3-phosphate dehydrogenase (GAPDH) blots, Western blotting with mouse monoclonal anti-polyubiquitin antibody detected a significant decrease in free ubiquitin accumulation in Δ*Mocsn6* and a significant increase in the Mcsn6c-H3 strain, compared with 70-15. Under CM nutrient conditions, free ubiquitin accumulation was decreased by 30.6% in Δ*Mocsn6* and increased by 72.2% in *Mocsn6c-H3*. After MG132 treatment, free ubiquitin accumulation was decreased by 10.9% in Δ*Mocsn6* and increased by 20.0% in *Mocsn6c-H3* (Fig. 10A and B).

To detect the ubiquitination level of MoAtg6, the MoAtg6-GFP vector was transformed into 70-15 and Δ*Mocsn6* mutant. Three of the 70-15 and three of the Δ*Mocsn6* mutant expressing MoAtg6-GFP protein were used as three independent replicates. Next, GFP beads were used to incubate protein extracts of 70-15 and Δ*Mocsn6* mutant with GFP-MoAtg6. Protein was collected and immunoblotted with the anti-polyubiquitin antibody that only recognizes polyubiquitinated ubiquitin chains. As expected, the ubiquitination level of MoAtg6 in the Δ*Mocsn6* mutant was significantly lower than that of 70-15, which was reduced by 36.3% in similar GFP-Atg6 Western blots (Fig. 10C and D). These data show that MoCsn6 plays an active role in regulating ubiquitination levels and that the ubiquitination level of MoAtg6 is affected by MoCsn6.

## DISCUSSION

Increasing evidence has shown that autophagy is essential for the virulence of many pathogenic fungi, such as *M. oryzae*, *Beauveria bassiana*, *Fusarium graminearum*,

*Botrytis cinerea*, and *Phytophthora sojae* (40–44). However, the regulatory mechanisms of autophagy in pathogenic fungi remain unclear. The Csn8 subunit has been reported to regulate the maturation of autophagosomes in mouse cardiomyocytes, but the specific mechanisms underlying the relationship between CSN and autophagy remain to be elucidated (28). Deletion of any CSN subunit is lethal for higher eukaryotes, and this makes fungi good models for studying the molecular mechanism of CSN. The biological functions of the MPN domain subunit Csn5 have been revealed in fungi such as *Saccharomyces. cerevisiae*, *B. bassiana*, and *Alternaria alternata* (8, 9, 45), but the functions of another MPN domain subunit, Csn6, have not yet been reported. In this study, we report the biological function of MoCsn6 in regulating pathogenicity and autophagy in *M. oryzae*. Furthermore, we preliminarily investigated the cross talk between MoCsn6-mediated MoAtg6 ubiquitination and autophagy.

In *B. bassiana*, BbCsn5 regulates the transcription of two key central developmental pathway (CDP) genes, *brl*A and *aba*A, to mediate the production of aerial conidiation (9). Deletion of *BbCSN5* significantly reduced conidia yield, hydrophobicity, and adhesion to the insect cuticle, which reduced pathogenicity. Furthermore, in *A. alternata*, the Δ*Aacsn5* mutant had a reduced growth rate and lost the ability to produce conidia compared with the wild-type strain (8). Deletion of *AaCSN5* significantly reduced necrotic lesions on citrus leaves. Many factors, including conidial germination and appressorium formation, are associated with the pathogenicity of *M. oryzae* (29, 46). Our study showed that MoCsn6 is necessary for the infection-related morphogenesis of *M. oryzae*. The Δ*Mocsn6* mutants were severely defective in vegetative hyphal growth, conidiation, germ tube germination, and appressorium formation, which ultimately resulted in the inability to produce robust infectious structures appressoria. The pleiotropic defect in the Δ*Mocsn6* mutant resulted in its inability to penetrate plant epidermal cells, which reduced its pathogenicity. To our knowledge, this is the first study to report that Csn6 is related to the virulence of plant-pathogenic fungi. In addition, appressorium formation is regulated by the MAPK pathway; abnormal Pmk1 phosphorylation leads to reduced or lost pathogenicity in *M. oryzae*. The phosphorylation level of Osm1 is related to the sensitivity of pathogenic fungi to hyperosmotic stress (47). In this study, MoCsn6 was found to be involved in the Osm1 and Pmk1 phosphorylation pathways. Compared with 70-15, the phosphorylation levels of Osm1 and Pmk1 were higher in the Δ*Mocsn6* mutant. This was consistent with the result that the Δ*Mocsn6* mutant had impaired appressorium formation and was more sensitive to hyperosmotic stress than 70-15. In addition, we identified CSN subunits in *M. oryzae* with mammalian homology. The MPN domain and PIC domain in the CSN subunit of *M. oryzae* correspond to those of mammals. The experimental results of the interaction between CSN subunits also prove that the COP9 signalosome is highly conservative.

CSN has multiple functions, including participating in the cell cycle and ubiquitin-dependent protein degradation processes, regulating cullin-RING ubiquitin ligases as holoenzymes, and serving as a docking platform for a series of kinases and other intracellular regulatory proteins (48, 49). In addition to functioning as a complex, each CSN subunit may have unique functions, and some individual CSN subunits are involved in different protein-protein interactions (50). In our study, the autophagy core protein hooks the MoCsn6 protein in rice blast cDNA libraries, and MoCsn6 interacts with MoAtg6 both *in vitro* and *in vivo*. To our knowledge, this is the first study to show that Csn6 is involved in the autophagy regulatory pathway.

The normal function of cells depends on intracellular proteins, and the quantity and function of cellular proteins rely on the dynamic balance of protein synthesis and degradation (12, 51). Therefore, the ubiquitin-proteasome system (UPS) and autophagy-lysosome system, which are responsible for degrading cellular proteins, are very important for cell signal transduction and function. The relationship between CSN and autophagy was first discovered in 2008. The results showed that *CSN2* knockdown in the human myeloid progenitor cell line K562 resulted in the accumulation of the autophagy marker LC-II and autophagosomes. The phenotype of K562 cells treated with autophagy inhibitors was consistent with *CSN2*

knockdown, suggesting that the phenotype of CSN2 knockdown was caused by autophagy inhibition (51). The Csn8 subunit has also been reported to regulate autophagosome maturation in mouse cardiomyocytes and mediate the ubiquitin-proteasome system and autophagy (52). Using mouse CR-CSN8$^{CKO}$ hearts (in which Csn8 was conditionally ablated in mouse cardiomyocytes) as a model, the level of the autophagy marker protein p62/SQSTM1 was significantly increased in CR-CSN8$^{CKO}$ cardiomyocytes (28). LC3-II protein, a marker of autophagy abundance, did not increase, the fusion between autophagosomes and lysosomes was impaired, and Rab7 (a small GTPase protein known to be critical for the fusion process and vesicle transport) was downregulated. These results suggested that Csn8 is required for autophagy in mouse cardiomyocytes. In *M. oryzae*, a key autophagy threshold is critical to pathogenicity, and autophagy that is too fast or slow significantly affects the virulence of pathogenic fungi (22, 53). MoAtg8 lipidation and GFP-MoAtg8 degradation are commonly used to measure autophagic flux (54). Our results indicated that, compared with wild-type 70-15, lipidated MoAtg8 accumulates more and GFP-MoAtg8 degrades faster in Δ*Mocsn6* mutants under both nutrient and nitrogen deficiency conditions. The autophagy level was higher in Δ*Mocsn6* mutants than in 70-15, and MoCsn6 negatively regulated autophagy in *M. oryzae*.

CSN plays an important regulatory role in the intracellular ubiquitination-deubiquitination pathway (48). In *Aspergillus nidulans*, six subunits of the COP9 signalosome interact with a novel ubiquitin-specific protease, UspA (Usp15 in *Homo sapiens*), to mediate fungal multicellular development (55). Intact CSN represses *AnUSPA* gene expression and reduces the number of ubiquitinated proteins during fungal development. In the pathogenic fungus *B. bassiana*, deletion of *BbCSN5* results in a 68% increase in intracellular ubiquitin accumulation and dysregulation of 18 ubiquitin-related genes, suggesting that Csn5 plays a role in balancing ubiquitin and deubiquitin (9). In our study, the deletion of *MoCSN6* resulted in decreased overall ubiquitination levels in cells, indicating that the intact CSN complex may be involved in this process through deneddylation in *M. oryzae*. Compared with the wild type 70-15, the ubiquitination of MoAtg6 was reduced in the Δ*Mocsn6* mutant. These results suggested that MoCsn6 is involved in the regulation of ubiquitination in *M. oryzae* and mediates the ubiquitination level of MoAtg6 by interacting with it.

Atg6 is a homolog of mammal Beclin1 and yeast Vps30. Previous studies have shown that Atg6 is an important component of the phosphatidylinositol 3 kinase complex and is essential for autophagy (26). The Beclin1 level is one of the key factors affecting autophagy induction, which is very important for reducing aggregated proteins. In a mouse model of Niemann-Pick C disease, increased Beclin1 expression mediates the upregulation of autophagy (56). In *Caenorhabditis elegans*, downregulation of Beclin1 inhibits autophagy (57). In this study, MoCsn6 interacted with MoAtg6, and the deletion of *MoCSN6* reduced the ubiquitination level of MoAtg6 and enhanced autophagy. These findings, combined with the results of previous studies, indicate that MoCsn6 interacts with MoAtg6 to mediate the ubiquitination of MoAtg6 and mediates autophagy by controlling the expression of MoAtg6.

In conclusion, we found that the COP9 signalosome subunit MoCsn6 is involved in autophagy regulation and characterized its biological function. MoCsn6 is essential for the vegetative growth, spore production, germ tube germination, appressorium formation, and pathogenesis of *M. oryzae*. MoCsn6 was required for the appropriate phosphorylation of Pmk1 and Osm1. In addition, MoCsn6 regulates the ubiquitination level of the autophagy core protein MoAtg6 by interacting with it; MoCsn6 also participates in the cross talk between the autophagy pathway and ubiquitination pathway. Previous studies of CSN have rarely addressed the question of whether and to what extent different CSN subunits work separately or in a complex. We have obtained the deletion mutants for other CSN subunits and will continue to explore this interesting question. In addition, future studies are needed to explore the precise molecular functions of CSN and its subunits in the autophagy regulation process, which is of great significance for the development of new treatment strategies for rice blast fungus.

## MATERIALS AND METHODS

**Strains, growth conditions, and quantitative real-time PCR.** We used 70-15 as the wild-type strain in this study. All fungal strains were grown in complete medium (CM) at 25°C for 8 to 10 days. To validate the sensitivity of the ΔMocsn6 mutant to hyperosmotic stresses, 70-15, ΔMocsn6 mutant, and Mocsn6c were cultured on CM with 0.8 M sorbitol, 0.4 M KCl, or 0.6 M NaCl. The assay was independently repeated three times. The relative growth rate was calculated using the following formula: (diameter of treated strain)/(diameter of untreated strain). The quantitative real-time PCR primers used in this experiment are shown in Table S2 in the supplemental material.

**Targeted gene deletion or complementation.** Based on the principle of homologous recombination, we constructed a gene deletion vector using the high-throughput gene knockout system designed by Jian-Ping Lu and inserted the native gene copy into the ΔMocsn6 mutant to obtain the complemented strain Mocsn6c (58). To knock out the target gene MoCSN, we performed the following experiments. The 1.5- to 2.0-kb upstream fragment (UF) and downstream fragment (DF) of MoCSN6 were amplified from the genomic DNA of the wild-type strain 70-15 using the primers Up-F/Up-R and Down-F/Down-R. A 1.3-kb hygromycin resistance gene (HPH) fragment was amplified from pCB1003 using primer pair HPH-F/HPH-R. The above three fragments were linked to the HindIII/XbaI-linearized vector pKO3A by ligase (Vazyme, P505-d1) (Fig. S2A). The knockout vector was transformed into the wild-type strain 70-15 by Agrobacterium tumefaciens-mediated transformation (ATMT) methods. The primers S-F/S-R and Long-HPH-R/Long-F were used to verify the long fragment and short fragment of the mutant (Fig. S2B). The copy number of the resistance gene HPH was then verified by quantitative real-time PCR, and the β-tubulin gene was used as a control (Fig. S2C). To construct the complement vector, we cloned the target gene sequence with an approximately 2.0-kb native promoter from the wild-type strain 70-15. The segment was linked to EcoRI/BamHI-linearized pKD5 by ligase. The vector was then transformed into the ΔMocsn6 mutant using the ATMT strategy. The primers used to amplify the above sequences are shown in Table S2.

**Fluorescence observation.** To observe the subcellular localization of MoCsn6, we first transformed the MoCsn6-GFP vector into the ΔMocsn6 mutant to obtain the complemented strain. The H$_2$B-mCherry (a nuclear localization marker) fusion protein was transformed into the complemented strain with MoCsn6-GFP. The conidia were washed from the fungal colony that had grown on the solid CM plate for 9 days. After centrifuging and washing three times, the conidia were inoculated on hydrophobic plastic film in a wet box at 25°C. The localization of red and green fluorescence in conidia and appressorium was observed under a fluorescence microscope.

**Phenotypic characterization.** To measure the vegetative growth rate and sporulation, 70-15, ΔMocsn6 mutant, and Mocsn6c were inoculated on 7-cm solid CM at 25°C for 9 days with a 16-h light, 8-h dark cycle. The colony diameter was measured to indicate the vegetative growth rate, the conidia on the colony were washed off with 3 mL of water, and the sporulation quantity was determined under an optical microscope. To observe germ tube germination and appressorium formation, conidia were collected from colonies of 70-15, ΔMocsn6 mutant, and complementary strain Mocsn6c, which were grown on CM plates for 9 days. After washing off residual nutrients and mycelium, the $5 \times 10^4$ mL$^{-1}$ conidial suspensions were inoculated on a hydrophobic plastic film and cultured in darkness at 25°C. The appressorium formation rate was then calculated under a light microscope 8 h and 24 h later. For the pathogenicity assay, mycelial plugs and $5 \times 10^4$ mL$^{-1}$ conidial suspensions were inoculated on detached leaves of barley, and the spots were observed 4 days later. Mycelial plugs were inoculated on wounded barley and rice leaves, and the spots were observed 3 days later. To evaluate the ability of appressorium to infect the plant surface, the leaves of barley inoculated with $5 \times 10^4$ mL$^{-1}$ conidial suspensions for 36 h were collected and decolorized in methanol. Appressorium infection on plant leaves was then analyzed under a light microscope.

**Western blot analysis.** To test phosphorylated Pmk1, mycelia were cultured in liquid yeast extract-glucose at 25°C at 150 rpm for 40 h. To test phosphorylated Osm1, mycelia were first cultured in liquid CM at 25°C at 150 rpm for 40 h. They were then divided into four parts, one of which was used as the sample for 0 h, and the other three mycelia were transferred to liquid CM containing 0.6 M NaCl at 25°C at 180 rpm for 30 min, 60 min, and 90 min. To detect overall ubiquitination, mycelia were first cultured in liquid CM at 25°C at 150 rpm for 40 h, and then they were divided into two parts, one of which was used as the sample for 0 h, and the other part was transferred to liquid CM supplemented with MG132 at 25°C at 180 rpm for 4 h. The antibodies or agents used in the above experiments were anti-GAPDH (Huabio, Hangzhou, China), anti-phospho-Osm1 (Cell Signaling Technology), anti-Pmk1 (Cell Signaling Technology), anti-phospho-Pmk1 (Santa Cruz Biotechnology), anti-polyubiquitin (Cell Signaling Technology), and MG-132 (Cell Signaling Technology).

**Autophagy assays.** To detect the degradation of Atg8, Atg8-GFP with the native promoter mentioned in previous studies was transformed into the wild-type strain 70-15 and the ΔMocsn6 mutant using the ATMT strategy (59). The 70-15 strain and ΔMocsn6 mutants expressing Atg8-GFP were cultured in liquid CM at 25°C at 150 rpm for 40 h. After that, mycelia were divided into three parts, one of which was used as the sample for 0 h, and the other two were transferred to liquid nitrogen starvation (SD-N) medium for 3 and 6 h to induce autophagy. The above mycelia at 0 h and 3 h were stained with 7-amino-4-chloromethylcoumarin (CMAC) for 5 min, and then green fluorescence and blue fluorescence were observed under a fluorescence microscope. To detect the transformation of Atg8 and Atg8-PE, wild-type 70-15 and ΔMocsn6 mutant were cultured in liquid CM at 25°C at 150 rpm for 40 h. After that, mycelia were divided into three parts, one of which was used as the sample for 0 h, and the other two were transferred to liquid SD-N medium for 3 and 6 h to induce autophagy. The antibodies or agents used in the above experiments were anti-GAPDH antibody (Huabio, Hangzhou, China), anti-GFP antibody (Huabio, Hangzhou, China), anti-Atg8 antibody (BML Beijing Biotech), Goat anti-Rabbit IgG (Huabio, Hangzhou, China), Goat anti-Mouse IgG (Huabio, Hangzhou, China), and CMAC. A 13.5% SDS-

PAGE with 6 M urea was used to separate Atg8 and Atg8-PE with similar molecular weights. Then, 12.5% SDS-PAGE without additional urea was used to detect the exogenous insertion protein GFP-Atg8/GFP.

**Yeast two-hybrid assays and coimmunoprecipitation assays.** The sequence of the target gene was amplified from full-length cDNA, and the amplification products were linked to the *Eco*RI/*Bam*HI-linearized AD (pGADT7) vector or BD (pGBKT7) vector. pGADT7-T and pGBKT7-53 were used as positive controls. The yeast two-hybrid assay was carried out according to the steps of the Matchmaker Gal4 two-hybrid system 3 (Clontech, USA). In coimmunoprecipitation assays, MoAtg6-GFP was constructed and transformed into the Δ*Mocsn6* mutant complemented with MoCsn6-3×Flag. MoCsn1-GFP, MoCsn3-GFP, MoCsn4-GFP, MoCsn5-GFP, and MoCsn7a-GFP were constructed and transformed into the 70-15 strain with MoCsn6-3×Flag. Flag antibody (Huabio, Hangzhou, China) and GFP antibody (Huabio, Hangzhou, China) were used to detect expression of the GFP label in transformants by Western blotting. Total protein was extracted with protein lysis buffer and then incubated with anti-GFP beads for 4 h. Next, the protein was washed twice with high salt and low salt and eluted with 200 $\mu$L of glycine elution buffer for Western blot analysis. The primers used are shown in Table S2.

## SUPPLEMENTAL MATERIAL

Supplemental material is available online only.

**SUPPLEMENTAL FILE 1**, PDF file, 1.2 MB.

## ACKNOWLEDGMENTS

This study was supported by the Key Research and Development Project of Zhejiang Province, China (2021C02010), the Special Project for the Selection and Breeding of New Agricultural Varieties in Zhejiang Province, China (2021C02064), and grants from the National Natural Science Foundation of China (32270201, 31972216 and 31970140).

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
