## [Reviewer comments · Microbiology Spectrum]

Microbiology Spectrum

A subunit of the COP9 signalosome, MoCsn6, is involved in fungal development, pathogenicity, and autophagy in rice blast fungus

Zi-Fang Shen, Lin Li, Jing-Yi Wang, Yun-Ran Zhang, Zi-He Wang, Shuang Liang, Xue-Ming Zhu, Jianping Lu, Fu-Cheng Lin, and Xiao-Hong Liu

Corresponding Author(s): Xiao-Hong Liu, Institute of Biotechnology

Review Timeline:

Submission Date:	May 31, 2022
Editorial Decision:	July 27, 2022
Revision Received:	September 10, 2022
Editorial Decision:	September 30, 2022
Revision Received:	October 2, 2022
Accepted:	October 24, 2022

Editor: Patricia Albuquerque

Reviewer(s): Disclosure of reviewer identity is with reference to reviewer comments included in decision letter(s). The following individuals involved in review of your submission have agreed to reveal their identity: Tong-Bao Liu (Reviewer #1)

Transaction Report:

DOI: <https://doi.org/10.1128/spectrum.02020-22>

July 27, 2022

Dr. Xiao-Hong Liu
Institute of Biotechnology
Hangzhou
China

Re: Spectrum02020-22 (A subunit of COP9 signalosome, MoCsn6, is involved in fungal development, pathogenicity, and autophagy in rice blast fungus)

Dear Dr. Xiao-Hong Liu:

Thank you for submitting your manuscript to Microbiology Spectrum. All the reviewers and I believe that your manuscript has interesting data, but needs improvement in the writing and a few other modifications. Therefore, I suggest that you make the suggested modifications and answer all the reviewer's questions. When submitting the revised version of your paper, please provide (1) point-by-point responses to the issues raised by the reviewers as file type "Response to Reviewers," not in your cover letter, and (2) a PDF file that indicates the changes from the original submission (by highlighting or underlining the changes) as file type "Marked Up Manuscript - For Review Only". Please use this link to submit your revised manuscript - we strongly recommend that you submit your paper within the next 60 days or reach out to me. Detailed instructions on submitting your revised paper are below.

Link Not Available

Sincerely,

Patricia Albuquerque

Journals Department
Reviewer comments:

Reviewer #1 (Comments for the Author):

The manuscript authored by Shen et al. identified MoCsn6, a subunit of COP9 signalosome, and characterized its biological roles in the rice blast fungus *M. oryzae*. Deletion of the MoCSN6 resulted in reduced hyphal growth, decreased sporulation and germ tube germination, and reduced pathogenicity in *M. oryzae*. The authors' further research showed that the Osm1 and Pmk1 phosphorylation pathways were also disrupted in the Δ Mocsn6 mutants. Interestingly, the authors found that MoCsn6 participates in the autophagy pathway by interacting with the autophagy core protein MoAtg6 and regulating the ubiquitination level of MoAtg6. Furthermore, the authors proved that MoCsn6 is involved in the negative regulation of autophagic activity. Based on the above research results, the authors claimed that the MoCsn6 plays a crucial role in regulating fungal development,

pathogenicity, and autophagy in *M. oryzae*.

I appreciated the authors' demanding work for systematic research on the MoCSN6 gene. The general research strategies and methodologies are basically sound, and the study is well executed. However, since the authors claimed that the MoCsn6 played an important role in autophagy, I wonder whether the authors examined the formation of autophagosomes in Δ Mocsn6 mutant strains?

Meanwhile, there are many writing problems in the manuscript. Please carefully read through the manuscript to check grammar and spelling errors. There are lots of grammar errors and several typos, including but not limited to the following:

- L15: replace "which affects a variety of" with "affecting various"
- L21: "reduced" should be "reduce"
- L24: remove "firstly"
- L35: remove the "," after "*M. oryzae*"
- L70: remove the "," after "structures"
- L105: replace the "," with ";"
- L106: replace "not necessary" with "unnecessary"
- L107: add a "," after "cellular function"
- L109: remove the "databases" after "NCBI"
- L110: add "a" preceding "conserved MPN domain" and "conserved PIC domain"
- L115: add a "," after "MoCsn5"
- L125: "wild type" should be "wild-type"
- L128: add "is" preceding "also"; add "the" preceding "cytoplasm"
- L132: add "the" preceding "target gene replacement method"
- L140: add "the" preceding "mutant"
- L141: "Figure 4B, C and D" should be "Figure 4B, C, and D"
- L144: add a "," after " Δ Mocsn6"
- L146: remove the "," after "conidiophores"
- L154: add a "," after " Δ Mocsn6 mutant"
- L155: replace "was" with "were"
- L169: add a "," after " Δ Mocsn6"
- L174: add a "," after " Δ Mocsn6"
- L200: add a "," after "vector"
- L201: remove the "," after "positive control"
- L205: add "it" preceding "was"
- L216: remove the "," after "ATG proteins"
- L230: remove the "," after " Δ Mocsn6"
- L231: "nutrient condition" should be "nutrient conditions"
- L232: add "the" preceding "cytoplasm"
- L233: add "the" preceding "vacuole"; replace the "," after "completely" with ";"
- L234: add "the" preceding "cytoplasm"
- L235: "nutrient condition" should be "nutrient conditions"
- L236: add "the" preceding "starvation"
- L242: add a "," after "70-15"
- L244: add a "," after "became weak"
- L266: "evidences" should be "evidence"
- L267: *Beauveria bassiana* is an entomogenous fungus
- L272: add "the" preceding "MPN"
- L273: "*Saccharomyces. Cerevisiae*" should be "*S. Cerevisiae*"; add a "," after "*B. bassiana*"
- L278: add a "," after "brlA"
- L279: add a "," after "hydrophobicity"
- L289: replace the "," with ";"
- L305: remove "cell" preceding "function"
- L333: "*M. oryzae*" should be italic and remove the "," after "*M. oryzae*"
- L351: change "to develop" to "in developing"
- L356: add a "," after "0.4 M KCl"
- L383: remove the "fluorescence" after red
- L391: add a "," after "Mocsn6c"
- L393: add "the" preceding "appressorium formation"
- L414: "at150" should be "at 150"
- L433: add a "," preceding "respectively"
- L573: "were" should be "was"
- L575: "subunts" should be "subunits"
- L580: ", " should be ", "
- L591: "antibody" should be "antibodies"

Reviewer #2 (Comments for the Author):

Generally the manuscript is well written and figures are well represented. I have some minor issues with the manuscript which think should be addressed .They are as follows:-

Line 133: The authors did not do southern blot to confirm the deletion mutants? Why?

Line 115: The authors did not validate the yeast two hybrid interactions for Figure 1 with either Coip, or BiFC techniques. Validation is required for these kind of assays.

Line 121: How about the localization of MoCsn6 in conidia and hypha? Why not display the pictures.

Line 143: Why do you think the colony morphology of MoCsn6 mutant is dark brown?

Line 159: why not include the conidial germination graph for the mutant at different time points?

Line 162: Did you assay for Pmk1 phosphorylation at appressorium stage or using Mycelium? Please indicate in the text.

Line 170: Did you test the pathogenicity on wounded barley/rice leaves? Do you think the pathogenicity will still be compromised?

Line 182: Do you have any experimental evidence to show why the MoCsn6 mutant is unable to proliferate /colonize the underlying host tissues?

Line 197: What is the control for this experiment, I mean what is the phosphorylation level of Csn6 during hypoosmotic stress as compared to guy11.It would be better to include untreated sample in the assay.

Reviewer #3 (Comments for the Author):

The paper by Shen and colleagues describes the involvement of the COP9 Signalosome, and in particular the 6th subunit of the complex, in the development, pathogenicity and autophagy in *Magnaporthe oryzae*. The paper describes a variety of physiological and pathogenic traits in fungus growth and development and points to a link between the CSN complex and pathogenicity through regulation of autophagy.

In general, the data shown in this paper are quite solid, however the link to autophagy is less established. The results showing that there is an increase in autophagic flux in the deletion strain are not very convincing.

One additional question, which is often not addressed in the study of the CSN, is whether and to what extent, the different CSN subunits work alone or in an assembled complex. Some attempt to address this question would be very meaningful in this paper. This could be achieved by comparing the pathogenicity of Δ MoCsn6 to that of another core subunit deletion mutant, IP experiments, such as for example evaluation of the putative interactions between other CSN subunits and Atg6, co-localization with other core CSN subunits, etc.

Additional remarks:

- Line 26 - Please indicate shortly what is the role of MoAtg8-PE in autophagy, same as mentioned in line 25 for MoAtg6 (core autophagy protein).
- Line 26-27, and within the text (Figure 9) - Why does nitrogen starvation causes opposite effects on MoAtg8-PE (accumulation) and GFP-MoAtg8 (rapid degradation).
- Line 28 and within the text - It is not clear how the two opposite effects on Atg8 and Atg6 indicate negative regulation of autophagy.
- Line 46 - If I am not wrong, a ninth subunit of the CSN was identified several years ago. It is worth mentioning, even if not present in *Magnaporthe oryzae*.
- Line 98-102 - What type of screen? Y2H? The description of the screen is vague, and "data not shown" is not sufficient. Please elaborate some more.
- Line 105-106 - "And this protein with 41.19% amino acid homology and 93% coverage compared with Csn6 of *N. crassa*, thus, we named the protein MoCsn6 (Figure 1B)." Please correct grammar.
- Line 106 - Please explain what is "coverage".
- Lines 115-120 - How do the dual interaction experiments found in Figure 2B agree with the known structure of the CSN complex?
- Line 118-120 - This sentence is not clear. How do the interactions between MoCsn5 and MoCsn6, MoCsn5 and MoCsn7a, MoCsn6 and MoCsn7a help CSN complete its biological functions?
- Line 152 and other places throughout the manuscript - conclusions need to be stated in a less definite manner, such as for example: "In conclusion, the colony morphology, vegetative growth, and conidiation of Δ Mocsn6 are severely impaired, suggesting / implying / indicating that MoCsn6 participates in the development of *M. oryzae*".
- Line 158 - What is hpi?
- Line 164 - Figure 5B shows only a mild increase in the levels of P-Pmk1 in the Δ MoCsn6 strain. Moreover, in order to say that there is significance increase, it is required to perform statistical tests based on several biological repeats.
- Line 164 -The increase in the level of P-Pmk1 in Δ Mocsn6 is very mild. This is not sufficient to state that this over-activation is a reason in failure to form appressorium in Δ Mocsn6. This assumption needs to be either toned down or validated by other means (activation of P-Pmk1 by chemicals? Overexpression of Pmk1? Other means?).
- Line 183 - Figure 6A middle panel, shows that inoculation of barley leaves with mycelial plugs from Δ Mocsn6 resulted in formation of few dark spots without serious lesions. This indicates that MoCsn6 is not entirely indispensable for pathogenicity. Please rephrase.

- Line 192 - Please indicate what is the relationship between hyper-osmotic stress and Osm1 MAPK signaling.
- Line 195 - This is not a contrast. In fig. 7C, it seems that the overall phosphorylation level of Osm1 in Δ Mocsn6 is higher than in 70-15, but it still goes up and down over time, as in the 70-15 strain.
- Line 195 - Fig. 7C - statistical tests are required to show whether there is a significance difference between 70-15 and Δ Mocsn6.
- Line 196-197 - Please tone down, results are not that clear.
- Figure 9A - Is the labeling of MoATG8 and MoAtg8-PE correct? Does MoATG8 migrate slower than MoATG8-PE in SDS-PAGE?
- Quantification (+ statistics from at least three repeats) are required for an accurate comparison of the degradation rates in 9A and 9B, as well as for the accumulation of the free GFP.
- Line 232 - It is not entirely clear what the authors mean here. The differences shown in Fig 9C between the two stains are mild. How is autophagy monitored in these images? Can there be any statistics for the puncta?
- Line 247 - In 9A, at 0h, the levels of MoAtg8 in the Δ MoCsn6 strain are higher than in 70-15, while in 9B, the levels of GFP-MoAtg8 in Δ MoCsn6 are lower than in 70-15. Can the authors explain this?
- Why is there no GFP-MoATG8-PE detected in 9B?
- Line 235-237 - Is it possible that the word "even" may not be in the correct context here? Weren't puncta observed in both strains after nutrient conditions?
- Line 255-259 - This paragraph is not written well. Please rephrase.
- Figure 10 - Why is the H3 strain used here and not in the other experiments? Please show these results in the GFP strain without the H3. Also, the labels for the different lanes in 10C are missing.
- What lanes in 10C were quantified in 10D?
- Line 333 - The decreased overall ubiquitination in cells of the Δ Mocsn6 strain indicates that the intact CSN complex is involved in the process, most probably though regulating the level of ubiquitination through deneddylation.
- Line 359 - I suggest to change the word "and" to "or".
- Line 361 - I may have missed this information but where was RT-PCR used in this experiment?

Staff Comments:

Preparing Revision Guidelines

Please return the manuscript within 60 days; if you cannot complete the modification within this time period, please contact me. If you do not wish to modify the manuscript and prefer to submit it to another journal, please notify me of your decision immediately so that the manuscript may be formally withdrawn from consideration by Microbiology Spectrum.

The authors have demonstrated that CSN6 is crucial in the fungal development and virulence of rice blast fungus. They have also shown that this gene negatively regulates autophagy by interacting with a core autophagy protein ATG6. Considering the crucial role of autophagy in fungal pathogenesis, this manuscript is of great interest to the broader scientific community and has highlighted important findings on the relationship between CSN complex and autophagy process. In addition the paper has identified Csn6 to act as negative regulator of Pmk1-MAPK and deletion of this protein resulted in overactivation of the Pmk1 pathway. As a result the appressorium formation was compromised in the mutant resulting to a defect in penetration of the underlying host tissues. Generally the manuscript is well written and figures are well represented. I have some minor issues with the manuscript which think should be addressed .They are as follows:-

Line 133: The authors did not do southern blot to confirm the deletion mutants? Why?

Line 115: The authors did not validate the yeast two hybrid interactions for Figure 1 with either Coip, or BiFC techniques. Validation is required for these kind of assays.

Line 121: How about the localization of MoCsn6 in conidia and hypha? Why not display the pictures.

Line 143: Why do you think the colony morphology of MoCsn6 mutant is dark brown?

Line 159: why not include the conidial germination graph for the mutant at different time points?

Line 162: Did you assay for Pmk1 phosphorylation at appressorium stage or using Mycelium? Please indicate in the text.

Line 170: Did you test the pathogenicity on wounded barley/rice leaves? Do you think the pathogenicity will still be compromised?

Line 182: Do you have any experimental evidence to show why the MoCsn6 mutant is unable to proliferate /colonize the underlying host tissues?

Line 197: What is the control for this experiment, I mean what is the phosphorylation level of Csn6 during hypoosmotic stress as compared to guy11.It would be better to include untreated sample in the assay.

Line 205: Which promoters was used in constructing the MoCsn6-GFP vector, as well as MoCsn6-flag vectors for co-immunoprecipitation assay?

Response to reviewer comments

Manuscript Number: Spectrum02020-22

Title: A subunit of COP9 signalosome, MoCsn6, is involved in fungal development, pathogenicity, and autophagy in rice blast fungus

The authors wish to thank the editors and reviewers for their time and efforts in reviewing our manuscript. The reviewers' comments and insight have been extremely helpful in improving the manuscript. Based on these comments, we have revised the manuscript by adding some experimental data and correcting some errors in the manuscript. The following list our responses to the comments on a point-to-point basis.

Reviewer #1 (Comments for the Author):

The manuscript authored by Shen et al. identified MoCsn6, a subunit of COP9 signalosome, and characterized its biological roles in the rice blast fungus *M. oryzae*. Deletion of the MoCSN6 resulted in reduced hyphal growth, decreased sporulation and germ tube germination, and reduced pathogenicity in *M. oryzae*. The authors' further research showed that the Osm1 and Pmk1 phosphorylation pathways were also disrupted in the $\Delta MoCsn6$ mutants. Interestingly, the authors found that MoCsn6 participates in the autophagy pathway by interacting with the autophagy core protein MoAtg6 and regulating the ubiquitination level of MoAtg6. Furthermore, the authors proved that MoCsn6 is involved in the negative regulation of autophagic activity. Based on the above research results, the authors claimed that the MoCsn6 plays a crucial role in regulating fungal development, pathogenicity, and autophagy in *M. oryzae*. I appreciated the authors' demanding work for systematic research on the *MoCSN6* gene. The general research strategies and methodologies are basically sound, and the study is well executed. However, since the authors claimed that the MoCsn6 played an important role in autophagy, I wonder whether the authors examined the formation of autophagosomes in $\Delta MoCsn6$ mutant strains?

Response: Thanks for your constructive comments. In *M. oryzae*, the amount of the autophagy reporter MoAtg8-PE is closely associated with the number of autophagosomes in the cell (Nair et al., 2012). The ratio between the amount of MoAtg8-PE and GAPDH can reflect the relative number of autophagosomes. We observed that the ratio between the amount of MoAtg8-PE and GAPDH in the $\Delta MoCsn6$ mutant was significantly higher than that of 70-15 by western blot, suggesting the number of autophagosomes increased significantly in the $\Delta MoCsn6$ mutant (Figure 9A). Furthermore, we observed autophagosomes in the mycelia of 70-15 and $\Delta MoCsn6$ mutant by light microscope. The results also showed that the number of autophagosomes in the $\Delta MoCsn6$ mutant was significantly more than that in 70-15 (Figure S4). (Nair, U., W.L. Yen, M. Mari, Y. Cao, Z. Xie, M. Baba, F. Reggiori, and D.J. Klionsky. 2012. A role for

Atg8-PE deconjugation in autophagosome biogenesis. *Autophagy*. 8:780-793.)

Meanwhile, there are many writing problems in the manuscript. Please carefully read through the manuscript to check grammar and spelling errors. There are lots of grammar errors and several typos, including but not limited to the following:

Response: Thank you for your careful review of our paper. We have carefully corrected grammatical and spelling errors in the manuscript, including but not limited to the errors you listed.

L15: replace "which affects a variety of" with "affecting various"

Response: Thanks a lot for your comments. We have modified it in our text.

L21: "reduced" should be "reduce"

Response: As suggested, we have modified it in our text.

L24: remove "firstly"

Response: Thanks. We have revised it in the text.

L35: remove the "," after "*M. oryzae*"

Response: We have revised this expression in our text according to the suggestion.

L70: remove the "," after "structures"

Response: We have removed it.

L105: replace the "," with ";"

Response: Thanks a lot for your comments. We have modified it in our text.

L106: replace "not necessary" with "unnecessary"

Response: As suggested, we have modified it in our text.

L107: add a "," after "cellular function"

Response: According to the suggestion, we have amended it in our text.

L109: remove the "databases" after "NCBI"

Response: Thanks. We have revised it in the text.

L110: add "a" preceding "conserved MPN domain" and "conserved PIC domain"

Response: Thank you for your suggestion, we have revised it in our text.

L115: add a "," after "MoCsn5"

Response: As suggested, we have added it to our text.

L125: "wild type" should be "wild-type"

Response: Thank you for your careful work. We have modified it in our text.

L128: add "is" preceding "also"; add "the" preceding "cytoplasm"

Response: We have added it to our text.

L132: add "the" preceding "target gene replacement method"

Response: According to the suggestion, we have amended it in our text.

L140: add "the" preceding "mutant"

Response: As suggested, we have added it to our text.

L141: "Figure 4B, C and D" should be "Figure 4B, C, and D"

Response: Thank you for your careful work. We have modified it in our text.

L144: add a "," after "*ΔMocsn6*"

Response: As suggested, we have added it to our text.

L146: remove the "," after "conidiophores"

Response: Thanks. We have revised it in the text.

L154: add a "," after "*ΔMocsn6* mutant"

Response: As suggested, we have added it to our text.

L155: replace "was" with "were"

Response: Thanks. We have revised it in the text.

L169: add a "," after "*ΔMocsn6*"

Response: Thanks. We have revised it in the text.

L174: add a "," after "*ΔMocsn6*"

Response: Thanks a lot for your comments. We have modified it in our text.

L200: add a "," after "vector"

Response: As suggested, we have added it to our text.

L201: remove the "," after "positive control"

Response: According to the comment, we have modified it in our text.

L205: add "it" preceding "was"

Response: We appreciate the positive suggestion and we have revised it in the text.

L216: remove the "," after "ATG proteins"

Response: Thanks. We have revised it in the text.

L230: remove the "," after "*ΔMocsn6*"

Response: As suggested, we have modified it in our text.

L231: "nutrient condition" should be "nutrient conditions"

Response: Thanks a lot for your comments. We have modified it in our text.

L232: add "the" preceding "cytoplasm"

Response: As suggested, we have modified it in our text.

L233: add "the" preceding "vacuole"; replace the "," after "completely" with ";"

Response: Thanks. We have revised it in the text.

L234: add "the" preceding "cytoplasm"

Response: We appreciate the positive suggestion and we have revised it in the text.

L235: "nutrient condition" should be "nutrient conditions"

Response: As suggested, we have modified it in our text.

L236: add "the" preceding "starvation"

Response: Thanks. We have revised it in the text.

L242: add a "," after "70-15"

Response: As suggested, we have modified it in our text.

L244: add a "," after "became weak"

Response: We have added it to our text.

L266: "evidences" should be "evidence"

Response: Thanks. We have revised it in the text.

L267: *Beauveria bassiana* is an entomogenous fungus

Response: We appreciate the positive suggestion and we have revised it in the text.

L272: add "the" preceding "MPN"

Response: We have added it to our text.

L273: "Saccharomyces. Cerevisiae" should be "S. Cerevisiae"; add a "," after "B. bassiana"

Response: As suggested, we have modified it in our text.

L278: add a "," after "brlA"

Response: Thanks. We have revised it in the text.

L279: add a "," after "hydrophobicity"

Response: We have added it to our text.

L289: replace the "," with ";"

Response: According to the suggestion, we have amended it in our text.

L305: remove "cell" preceding "function"

Response: Thanks. We have revised it in the text.

L333: "M. oryzae" should be italic and remove the "," after "*M. oryzae*"

Response: Thanks. We have revised it in the text.

L351: change "to develop" to "in developing"

Response: Thanks. We have revised it in the text.

L356: add a "," after "0.4 M KCl"

Response: We have added it to our text.

L383: remove the "fluorescence" after red

Response: According to the suggestion, we have amended it in our text.

L391: add a "," after "*Mocsn6c*"

Response: Thank you. We have revised it in the text.

L393: add "the" preceding "appressorium formation"

Response: Thanks. We have revised it in the text.

L414: "at150" should be "at 150"

Response: We have revised it in the text.

L433: add a "," preceding "respectively"

Response: According to the suggestion, we have amended it in our text.

L573: "were" should be "was"

Response: Thanks a lot for your comments. We have modified it in our text.

L575: "subunts" should be "subunits"

Response: Thanks. We have revised it in the text.

L580: ", " should be ","

Response: We have revised it in the text.

L591: "antibody" should be "antibodies"

Response: Thanks a lot for your comments. We have modified it in our text.

Reviewer #2 (Comments for the Author):

Generally the manuscript is well written and figures are well represented. I have some minor issues with the manuscript which think should be addressed. They are as follows:

Line 133: The authors did not do southern blot to confirm the deletion mutants? Why?

Response: Yes, we did not do Southern blot to confirm the deletion mutants, because we believed that two rounds of PCRs (one for deletion of the targeted gene, and another for the correct insertion of the deletion construct into the targeted site of the genome) combined with quantitative PCR (qPCR) (for off-target effects) can completely replace Southern hybridization to achieve the same effect (Figure S2). By selecting appropriate enzymes and probes, Southern hybridization can confirm whether the targeted gene knockout in the mutant and whether the resistance gene is inserted at multiple sites (off-target effects).

But Southern hybridization cannot verify point mutation in the mutant. Quantitative PCR has been used to determine the copy number of genes in the genome for a long time and is widely used in transgenic animals and plants (Ingham et al., 2001; Ginzinger et al., 2002; Ballester et al., 2004; Bubner et al., 2004; Shepherd et al., 2009; Giancaspro et al., 2017). In fungi, the use of qPCR to determine the copy number of genes in the genome has also been proposed (Peter et al., 2008; Dong et al., 2017). We have systematically described the process of a high-throughput gene knockout technology of *M. oryzae*, and compared the results between Southern hybridization, PCR, and qPCR for dozens of mutants (Lu et al., 2014; Cao et al., 2016; Cao et al., 2018; etc.). The results of Southern hybridization and qPCR are exactly the same. The deficiency of Southern hybridization also exists in qPCR, which also cannot identify random point mutations in mutants. Therefore, it is necessary to perform gene complementation for the obtained mutants, and observe whether the key mutant phenotypes are restored in complementation strains to determine whether the phenotypes of the mutants are caused by the deletion of the target gene. We have done complementation on knockout mutants. The vegetative growth, conidiation, appressorium formation, and pathogenicity of the complemented strains are rescued (Figure 4-6). Therefore, the mutants have no off-target and gene point mutation phenomenon, and the mutant phenotypes are caused by gene deletion.

References:

- Peter S. Solomon, Simon V.S. Ipcho, James K. Hane, Kar-Chun Tan, Richard P. Oliver, 2008. A quantitative PCR approach to determine gene copy number. *Fungal Genetics Reports* #55:5-8. (*Fungal Genetics Reports* #55 (fgsc.net))
- Wei-Xia Dong, Ming-Guang Feng, Sheng-Hua Ying, 2017, Use of quantitative PCR technique for determining gene copy number in the genome of *Beauveria bassiana* transformant. *Journal of Asia-Pacific Entomology*. 20(1), 57-59. <https://doi.org/10.1016/j.aspen.2016.12.001>
- Shepherd CT, Moran Lauter AN, Scott MP. Determination of transgene copy number by real-time quantitative PCR. *Methods Mol Biol*. 2009;526:129-34. doi: 10.1007/978-1-59745-494-0_11.
- Bubner B, Baldwin IT. Use of real-time PCR for determining copy number and zygosity in transgenic plants. *Plant Cell Rep*. 2004 Nov;23(5):263-71. doi: 10.1007/s00299-004-0859-y. Epub 2004 Sep 11. PMID: 15368076.
- Giancaspro A, Gadaleta A, Blanco A. Real-Time PCR for the Detection of Precise Transgene Copy Number in Wheat. *Methods Mol Biol*. 2017;1679:251-257. doi: 10.1007/978-1-4939-7337-8_15. PMID: 28913805.
- Ginzinger DG. Gene quantification using real-time quantitative PCR: an emerging technology hits the mainstream. *Exp Hematol*. 2002 Jun;30(6):503-12. doi: 10.1016/s0301-472x(02)00806-8. PMID: 12063017.
- Ingham DJ, Beer S, Money S, Hansen G. Quantitative real-time PCR assay for determining transgene copy number in transformed plants. *Biotechniques*. 2001 Jul;31(1):132-4, 136-40. doi: 10.2144/01311rr04. PMID: 11464506.
- Ballester M, Castelló A, Ibáñez E, Sánchez A, Folch JM. Real-time quantitative PCR-based system for determining transgene copy number in transgenic animals. *Biotechniques*. 2004 Oct;37(4):610-3. doi:

10.2144/04374ST06. PMID: 15517974.

Line 115: The authors did not validate the yeast two-hybrid interactions for Figure 1 with either Coip, or BiFC techniques. Validation is required for these kind of assays.

Response: Thanks for your constructive comments. We identified that the putative protein encoded by MGG_01432, which is associated with autophagy, is a subunit of the COP9 signalosome (CSN) and named it MoCsn6. Although CSN has been identified and studied in mammals and yeast, it has not been identified in *M. oryzae*. Here, we homologous aligned the yeast and mammalian COP9 subunits in *M. oryzae*, and identified the interaction between the subunits by the yeast two-hybrid assay, which resulted in evidence that these proteins are COP9 subunits. In this manuscript, we focus on the biological function of the subunit MoCsn6. To further clarify the relationship between MoCsn6 and other subunits of the COP9 signalosome, we performed the Co-IP assays in *M. oryzae in vivo*. The Co-IP results confirmed that MoCsn6 interacted with MoCsn1, MoCsn3, MoCsn4, MoCsn5, and MoCsn7a in rice blast. Co-IP evidence has been added in Figure S1.

Line 121: How about the localization of MoCsn6 in conidia and hypha? Why not display the pictures.

Response: Thanks for your suggestion. The localization of MoCsn6 in conidia has been shown in Figure 3A (the picture marked 0h on the left). In addition, according to your suggestion, the localization of MoCsn6 in hypha was observed (Figure 3B). Consistent with its localization in conidia and appressorium, MoCsn6 was also localized in the nucleus and cytoplasm of the hyphae.

Line 143: Why do you think the colony morphology of MoCsn6 mutant is dark brown?

Response: The wild-type strain 70-15 was used in this study. As shown in Figure 4A, the 70-15 strains grown on CM and MM solid medium for about 10 days were observed from the bottom, showing light color. However, we found that under the same growth conditions, the color of the bottom of the mutant colony was significantly different from that of 70-15, especially the centre of the colony showed a darker brown color than 70-15. In addition, as shown in Figure 4B, observation of the lateral sections of the colonies also clearly revealed that the color of the mutant mycelium was darker than that of the 70-15 and complemented strains, showing a dark brown color.

Line 159: why not include the conidial germination graph for the mutant at different time points?

Response: Thanks a lot for your comments. As your suggestion, we observed germ tube germination and appressorium formation of 70-15, $\Delta Mocsn6$ mutant, and *Mocsn6c* at 8 hours and 24 hours, and added pictures to Figure 5. In addition, the description of the results was added to the manuscript on line 156-159 (However, the appressorium formation rate was more than 90% in 70-15 and *Mocsn6c* at 8 hpi and 24 hpi, while only 4.8% in $\Delta Mocsn6$ mutant at 8 hpi and 7.4% at 24 hpi, and the conidial germination

rate of $\Delta Mocsn6$ was only 22.8% at 8 hpi and 30.1% at 24 hpi (Figure 5A, B and C)).

Line 162: Did you assay for Pmk1 phosphorylation at appressorium stage or using Mycelium? Please indicate in the text.

Response: Mycelium was used to perform Pmk1 phosphorylation assays, and we have refined the relevant statements in the text.

Line 170: Did you test the pathogenicity on wounded barley/rice leaves? Do you think the pathogenicity will still be compromised?

Response: According to your suggestions, we inoculated the wounded rice and barely leaves with mycelial plugs of three strains (70-15, $\Delta Mocsn6$ mutant, $\Delta Mocsn6c$) for 3 days, respectively. 70-15 and $\Delta Mocsn6c$ caused severe lesions, while the $\Delta Mocsn6$ mutant resulted in small disease lesions (Figure S3A and B). However, the pathogenicity of the $\Delta Mocsn6$ mutant partially recovered compared with inoculation on uninjured barley leaves when inoculating on wounded barley leaves (Figure 6A and B, S3A and B).

Line 182: Do you have any experimental evidence to show why the MoCsn6 mutant is unable to proliferate /colonize the underlying host tissues?

Response: The three-cell conidia germinate under appropriate environmental conditions to produce germ tubes, which then expand to form appressorium, a powerful weapon of *M. oryzae*. The mature appressorium punctures the host epidermal cells through extreme mechanical pressure, and then produces primary and secondary infecting hyphae, which colonize the host tissues and eventually lead to a rice blast outbreak. Combined with the pathogenic process of *M. oryzae* and our experimental results, we believe that there are three main reasons why the mutants cannot colonize the host tissues. First, the sporulation of the $\Delta Mocsn6$ mutant was significantly less than that of 70-15 (Figure 4). Second, the $\Delta Mocsn6$ mutant had low rates of germ tube germination and appressorium formation, and could not form sufficient numbers of appressorium with penetrating function (Figure 5). In addition, as shown in Figure 6, although the mutant's appressorium could produce primary infecting hyphae, it grew slowly, and was difficult to produce enough secondary infecting hyphae. This may be caused by the loss of pleiotropic function in growth, autophagy, and ubiquitination of the $\Delta Mocsn6$ mutant.

Line 197: What is the control for this experiment, I mean what is the phosphorylation level of Csn6 during hypoosmotic stress as compared to guy11. It would be better to include untreated sample in the assay.

Response: We used 70-15 as the wild-type strain in this study. In *M. oryzae*, the phosphorylation level of Osm1 changes in response to hyperosmotic stress, and the response mode has been previously studied

(Shi et al., 2019; Cai et al., 2022). Without adding NaCl, the phosphorylation of Osm1 was very low in wild-type strains 70-15/Guy11. When treated with NaCl, the phosphorylation level of Osm1 increased in the first 30 min and then gradually decreased (Sun et al., 2022). As shown in Figure 7C, Osm1 phosphorylation was low in both 70-15 and $\Delta MoCsn6$ mutants without 0.6 M NaCl treatment (lanes 1 and 5). When treated with 0.6 M NaCl, the phosphorylation level of Osm1 was increased in 70-15 and $\Delta MoCsn6$ in the first 30 min and then decreased later (lanes 2-4 and 6-8). It is worth noting that the phosphorylation levels of Osm1 in $\Delta MoCsn6$ were always higher than those in WT at all times.

(Cai, Y.Y., Wang, J.Y., Wu, X.Y., Liang, S., Zhu, X.M., Li, L. et al. (2022) MoOpy2 is essential for fungal development, pathogenicity, and autophagy in *Magnaporthe oryzae*. *Environ Microbiol* 24: 1653-1671.

Shi, H.B., Chen, N., Zhu, X.M., Su, Z.Z., Wang, J.Y., Lu, J.P. et al. (2019) The casein kinase MoYck1 regulates development, autophagy, and virulence in the rice blast fungus. *Virulence* 10: 719-733.

Sun, L., Qian, H., Wu, M., Zhao, W., Liu, M., Wei, Y. et al. (2022) A Subunit of ESCRT-III, MoIst1, Is Involved in Fungal Development, Pathogenicity, and Autophagy in *Magnaporthe oryzae*. *Front Plant Sci* 13: 845139.)

Line 205: Which promoters was used in constructing the MoCsn6-GFP vector, as well as MoCsn6-flag vectors for co-immunoprecipitation assay?

Response: The native promoter was used to construct the MoCsn6-Flag vector for the co-immunoprecipitation assay (Figure 8B). The native promoter was used to construct the MoCsn6-GFP vector for checking the localization of MoCsn6 (Figure 3).

Reviewer #3 (Comments for the Author):

The paper by Shen and colleagues describes the involvement of the COP9 Signalosome, and in particular the 6th subunit of the complex, in the development, pathogenicity and autophagy in *Magnaporthe oryzae*. The paper describes a variety of physiological and pathogenic traits in fungus growth and development and points to a link between the CSN complex and pathogenicity through regulation of autophagy.

In general, the data shown in this paper are quite solid, however the link to autophagy is less established. The results showing that there is an increase in autophagic flux in the deletion strain are not very convincing.

One additional question, which is often not addressed in the study of the CSN, is whether and to what extent, the different CSN subunits work alone or in an assembled complex. Some attempt to address this question would be very meaningful in this paper. This could be achieved by comparing the pathogenicity

of $\Delta MoCSN6$ to that of another core subunit deletion mutant, IP experiments, such as for example evaluation of the putative interactions between other CSN subunits and Atg6, co-localization with other core CSN subunits, etc.

Response: Thanks for your constructive comments. We also think that the question of whether the different CSN subunits work individually or in a combined complex is very interesting and worth investigating. Therefore, we tried to knock out the 7 CSN subunits identified in the rice blast fungus. However, only $\Delta MoCSN6$ mutant strain was obtained. Possibly because of the important function of CSN subunits in the rice blast fungus, the genes encoding other subunits failed to be knocked out. Our future research will continue to focus on this issue and try to explore it through other methods.

Additional remarks:

- Line 26 - Please indicate shortly what is the role of MoAtg8-PE in autophagy, same as mentioned in line 25 for MoAtg6 (core autophagy protein).

Response: Thanks for your suggestion. We have added relevant expressions.

Line 26-28: delete “Deletion of *MoCSN6* resulted in abnormal accumulation of MoAtg8-PE and rapid degradation of GFP-MoAtg8 in response to nitrogen starvation, suggesting that MoCsn6 is involved in the negative regulation of autophagic activity.”

Add “Deletion of *MoCSN6* resulted in rapid lipidation of MoAtg8 and degradation of the autophagic marker protein GFP-MoAtg8 under nutrient and starvation conditions, suggesting that MoCsn6 negatively regulates of autophagic activity.”

Line 225-227: add “Atg8-PE is the autophagy reporter that is reliably associated with completed autophagosomes. Changes in the amount of Atg8-PE are closely related to changes in the number of autophagosomes present in cells.”

- Line 26-27, and within the text (Figure 9) - Why does nitrogen starvation causes opposite effects on MoAtg8-PE (accumulation) and GFP-MoAtg8 (rapid degradation).

Response: The accumulation of MoAtg8-PE and the rapid degradation of GFP-MoAtg8 are not contradictory since they characterize different processes. Changes in MoAtg8-PE content were closely related to changes in the number of autophagosomes. In Figure 9A, the stronger the band of MoAtg8-PE, the greater the number of autophagosomes. The degradation of GFP-MoAtg8 indicates the rate of autophagosome fusion with the vacuole. In Figure 9B, the higher the value of GFP/ (GFP+ GFP-ATG8), the faster the autophagy rate.

- Line 28 and within the text - It is not clear how the two opposite effects on Atg8 and Atg6 indicate negative regulation of autophagy.

Response: The results on Atg8 and Atg6 are not the opposite. Atg8 is a marker protein for autophagy. The results in Figure 9 showed that autophagy was enhanced in $\Delta MoCsn6$ mutant, which proves that MoCsn6 was involved in the negative regulation of autophagy in *M. oryzae*. Combining the results in Figure 8 and Figure 10, we believe that MoCsn6 regulates the ubiquitination level of autophagy core protein MoAtg6 by interacting with it and participating in the cross-talk between the autophagy pathway and ubiquitination pathway. Less MoAtg6 was degraded by ubiquitination in the $\Delta MoCsn6$ mutant than in 70-15, which may lead to higher levels of MoAtg6 involved in autophagy regulation, resulting in enhanced levels of autophagy.

- Line 46 - If I am not wrong, a ninth subunit of the CSN was identified several years ago. It is worth mentioning, even if not present in *Magnaporthe oryzae*.

Response: Thanks a lot for your comments. We have added relevant content to the manuscript.

Line 58-62: add "The recently identified ninth subunit of CSN, CSN9, contributing to the steric regulation CRLs by reducing the affinity of CSN-CRL interactions, but is not necessary for the assembly and catalytic activity of CSN. CSN9 stoichiometrically complexes with CSN1-8 to form a nine-membered noncanonical CSN complex (also known as CSN9-bound CSN)."

- Line 98-102 - What type of screen? Y2H? The description of the screen is vague, and "data not shown" is not sufficient. Please elaborate some more.

Response: The autophagy core protein Atg6 was used as bait protein to screen the proteins interacting with Atg6 in the yeast AD library expressing all the proteins of *M. oryzae*. A total of 109 proteins were screened, and the gene encoding the corresponding protein was knocked out by the high-throughput gene knockout method, and a total of 76 deletion mutants were obtained. Data are presented in Supplementary Table 1.

- Line 105-106 - "And this protein with 41.19% amino acid homology and 93% coverage compared with Csn6 of *N. crassa*, thus, we named the protein MoCsn6 (Figure 1B)." Please correct grammar.

Response: Thank you. We have made changes to the text.

- Line 106 - Please explain what is "coverage".

Response: The "coverage" is a parameter of the sequence alignment result, which means the coverage of the submitted sequence relative to the target sequence, that is, the proportion of the submitted sequence participating in the alignment in NCBI. The "93% coverage" indicates that 93% of the amino acids in the sequence are involved in the alignment.

- Lines 115-120 - How do the dual interaction experiments found in Figure 2B agree with the known structure of the CSN complex?

Response: The amino acid sequences encoding mammalian CSN subunits were used for alignment in the NCBI databases, and the obtained CSN proteins in *M. oryzae* were used for the yeast two-hybrid assay in Figure 2B. Then, the genes encoding the CSN proteins of *M. oryzae* were searched in the UniProt databases to obtain the relevant information of the CSN structure.

- Line 118-120 - This sentence is not clear. How do the interactions between MoCsn5 and MoCsn6, MoCsn5 and MoCsn7a, MoCsn6 and MoCsn7a help CSN complete its biological functions?

Response: Thanks. This sentence was rephrased.

- Line 152 and other places throughout the manuscript - conclusions need to be stated in a less definite manner, such as for example: "In conclusion, the colony morphology, vegetative growth, and conidiation of $\Delta Mocsn6$ are severely impaired, suggesting / implying / indicating that MoCsn6 participates in the development of *M. oryzae*".

Response: Thank you for pointing this out, we have revised the relevant expressions in the manuscript.

- Line 158 - What is hpi?

Response: " 8 hpi " means "8 hours post-inoculation "; " 24 hpi " means "24 hours post-inoculation ". We have revised it in the text.

- Line 164 - Figure 5B shows only a mild increase in the levels of P-Pmk1 in the $\Delta Mocsn6$ strain. Moreover, in order to say that there is significance increase, it is required to perform statistical tests based on several biological repeats.

Response: Thanks for your constructive comments. We have adjusted relevant expressions. Three independent repeat experiments were carried out to detect the phosphorylation degree of Pmk1, and the results of the three experiments were consistent, the phosphorylation level of Pmk1 in $\Delta Mocsn6$ mutant was higher than 70-15. Figure 5D shows the result of one experiment, so no statistical analysis was performed.

- Line 164 -The increase in the level of P-Pmk1 in $\Delta Mocsn6$ is very mild. This is not sufficient to state that this over-activation is a reason in failure to form appressorium in $\Delta Mocsn6$. This assumption needs to be either toned down or validated by other means (activation of P-Pmk1 by chemicals? Overexpression of Pmk1? Other means?).

Response: Thanks for your constructive comments. The degree of Pmk1 phosphorylation in the $\Delta Mocsn6$ mutant was higher than 70-15 in three independent replicates, so we think that the change in the Pmk1 phosphorylation level is one of the reasons for the failure to form appressorium. And we are also unable to provide new evidence that MoCsn6 is involved in the phosphorylation of Pmk1. Therefore, we modified our conclusion as abnormal phosphorylation of Pmk1 is one of the reasons why $\Delta Mocsn6$ failed to form appressorium. And we have adjusted our conclusion.

- Line 183 - Figure 6A middle panel, shows that inoculation of barley leaves with mycelial plugs from $\Delta MoCsn6$ resulted in formation of few dark spots without serious lesions. This indicates that MoCsn6 is not entirely indispensable for pathogenicity. Please rephrase.

Response: Thanks. This sentence was rephrased according to the suggestion.

- Line 192 - Please indicate what is the relationship between hyper-osmotic stress and Osm1 MAPK signaling.

Response: Thanks. We have added relevant expressions.

Lin199-201: add “In *M. oryzae*, the MAPK Osm1-mediated signaling pathway is responsible for regulating the hypertonic stress response. To verify whether MoCsn6 regulates this pathway by affecting the phosphorylation of Osm1, we further examined the Osm1 MAPK signaling pathway.”

- Line 195 - This is not a contrast. In fig. 7C, it seems that the overall phosphorylation level of Osm1 in $\Delta MoCsn6$ is higher than in 70-15, but it still goes up and down over time, as in the 70-15 strain.

Response: In *M. oryzae*, the phosphorylation level of Osm1 changes in response to hyperosmotic stress, and the response mode has been previously studied (Shi et al., 2019; Cai et al., 2022). Without adding NaCl, the phosphorylation of Osm1 was very low in wild-type strains 70-15/Guy11. When treated with NaCl, the phosphorylation level of Osm1 increased in the first 30 min and then gradually decreased (Sun et al., 2022). As shown in Figure 7C, Osm1 phosphorylation was low in both 70-15 and $\Delta MoCsn6$ mutants without 0.6 M NaCl treatment (lanes 1 and 5). When treated with 0.6 M NaCl, the phosphorylation level of Osm1 was increased in 70-15 and $\Delta MoCsn6$ in the first 30 min and then decreased later (lanes 2-4 and 6-8). It is worth noting that the phosphorylation levels of Osm1 in $\Delta MoCsn6$ were always higher than those in WT at all times, indicating that MoIst1 is involved in controlling the phosphorylation of the Osm1 kinase to adapt to hyperosmotic stress.

(Cai, Y.Y., Wang, J.Y., Wu, X.Y., Liang, S., Zhu, X.M., Li, L. et al. (2022) MoOpy2 is essential for fungal development, pathogenicity, and autophagy in *Magnaporthe oryzae*. *Environ Microbiol* 24: 1653-1671.

Shi, H.B., Chen, N., Zhu, X.M., Su, Z.Z., Wang, J.Y., Lu, J.P. et al. (2019) The casein kinase MoYck1 regulates development, autophagy, and virulence in the rice blast fungus. *Virulence* 10: 719-733.

Sun, L., Qian, H., Wu, M., Zhao, W., Liu, M., Wei, Y. et al. (2022) A Subunit of ESCRT-III, MoIst1, Is Involved in Fungal Development, Pathogenicity, and Autophagy in *Magnaporthe oryzae*. *Front Plant Sci* 13: 845139.)

- Line 195 - Fig. 7C - statistical tests are required to show whether there is a significance difference between 70-15 and $\Delta MoCsn6$.

Response: Thanks a lot for your comments. Three independent repeat experiments were carried out to

detect the phosphorylation degree of Osm1, and the results were consistent. Figure 7C shows the results of one experiment. Figure 7D is only a more intuitive display of Osm1 phosphorylation in Figure 7C, similar to the degradation rate of autophagy in Figure 9B (Figure 9B is in the form of data and Figure 7D is in the form of a column diagram), so no statistical analysis was performed. We changed the column diagram in Figure 7D to a more visual line chart.

- Line 196-197 - Please tone down, results are not that clear.

Response: Thanks a lot for your comments. We have revised the relevant expressions.

Line 205-207: delete “The above results suggest that MoCsn6 plays an important role in regulating the Osm1 phosphorylation level in response to hyperosmotic stress.” Add “The above results suggest that MoCsn6 is involved in controlling the Osm1 phosphorylation level in response to hyperosmotic stress.”

- Figure 9A - Is the labeling of MoAtg8 and MoAtg8-PE correct? Does MoATG8 migrate slower than MoATG8-PE in SDS-PAGE?

Response: Yes, the label of MoAtg8 and MoAtg8-PE is correct. MoAtg8 migrates slower than MoAtg8-PE.

- Quantification (+ statistics from at least three repeats) are required for an accurate comparison of the degradation rates in 9A and 9B, as well as for the accumulation of the free GFP.

Response: Thanks for your suggestion. Three independent repeat experiments were carried out to detect the lipidation of MoAtg8 (Figure 9A) and degradation of GFP-MoAtg8 (Figure 9B). Figure 9A and B show the results of one experiment. Due to the different backgrounds of western blot, no statistical analysis of data was carried out, but the results of the three experiments were consistent. The Atg8-PE band of the mutant was stronger than that of 70-15, and the degradation rate of GFP-Atg8 of the mutant was faster than that of 70-15.

- Line 232 - It is not entirely clear what the authors mean here. The differences shown in Fig 9C between the two stains are mild. How is autophagy monitored in these images? Can there be any statistics for the

puncta?

Response: The cellular localization pattern of Atg8 is a reliable marker for autophagy in a variety of organisms. GFP-Atg8 in undegraded autophagosomes was observed as green fluorescent dots, while degraded autophagosomes showed fluorescence in the vacuole. In Figure 9C, vesicles are marked in blue by CMAC. Under CM conditions, green fluorescent dots in 70-15 were significantly more than those in the mutant, indicating that most of the autophagosomes in 70-15 remained undegraded (corresponding to lane 1 in Figure 9B, the GFP- Atg8 band is strong), while most of the autophagosomes in the mutants have been degraded (corresponding to lane 4 in Figure 9B, the GFP-Atg8 band is weak). After SD-N induction for 3 hours, the fluorescence in the mutants was all in the vacuole, and there was no visible punctate fluorescence, but some punctate fluorescence was still visible in 70-15. Figure 9C is a different representation of the results in Figure 9B, and the results of the three independent repeated experiments were consistent, so no statistical analysis of the data was performed.

- Line 247 - In 9A, at 0h, the levels of MoAtg8 in the $\Delta Mocsn6$ strain are higher than in 70-15, while in 9B, the levels of GFP-MoAtg8 in $\Delta Mocsn6$ are lower than in 70-15. Can the authors explain this?

Response: Since the endogenous Atg8/Atg8-PE was detected in Figure 9A, the degradation rate of the exogenous inserted protein GFP-Atg8 was detected in Figure 9B. In Figure 9A, Atg8-antibody was used to directly detect the turnover of endogenous Atg8/Atg8-phosphatidylethanolamine (Atg8-PE) to assess the number of autophagosomes. In Figure 9B, exogenous GFP-MoAtg8 fusion with native promoter vector was transformed into 70-15 and $\Delta Mocsn6$ to examine autophagosome degradation. As shown in Figure 9B, the value of GFP/(GFP+GFP-MoAtg8) in $\Delta Mocsn6$ was higher than that in 70-15, indicating that the degradation rate of autophagosome in $\Delta Mocsn6$ was faster than that in 70-15.

- Why is there no GFP-MoAtg8-PE detected in 9B?

Response: The Atg8-antibody was used to detect endogenous Atg8/Atg8-PE in Figure 9A. 13.5% SDS-PAGE with 6M urea was used to separate Atg8 and Atg8-PE with similar molecular weight. The GFP-antibody and 12.5% SDS-PAGE without additional urea were used to detect the exogenous insertion protein GFP-Atg8/GFP in Figure 9B, so there was no GFP-MoAtg8-PE detected.

- Line 235-237 - Is it possible that the word "even" may not be in the correct context here? Weren't puncta observed in both strains after nutrient conditions?

Response: Thanks. This sentence was rephrased.

- Line 255-259 - This paragraph is not written well. Please rephrase.

Response: Thanks. This sentence was rephrased according to the suggestion.

- Figure 10 - Why is the H3 strain used here and not in the other experiments? Please show these results

in the GFP strain without the H3. Also, the labels for the different lanes in 10C are missing.

Response: The strain with strong promoter H3 was used to detect the effect of high MoCsn6 expression on the overall ubiquitination level, so the H3 strain was used only in this assay. Strains 70-15 showed no H3-expressed MoCsn6. Compared with 70-15, the overall ubiquitination level was significantly reduced in the $\Delta Mocsn6$ mutant and significantly increased in the H3 strain. We believe that these results are sufficient to prove that MoCsn6 is involved in the positive regulation of the overall ubiquitination level.

To detect the ubiquitination level of MoAtg6, the MoAtg6-GFP vector was transformed into 70-15 and $\Delta Mocsn6$ mutant. Three of the 70-15 and three of the $\Delta Mocsn6$ mutant expressing MoAtg6-GFP protein were used as three independent replicates. As suggested, we have added labels above the lanes.

- What lanes in 10C were quantified in 10D?

Response: Figure 10D is a statistical analysis of the three experimental results in Figure 10C.

- Line 333 - The decreased overall ubiquitination in cells of the $\Delta Mocsn6$ strain indicates that the intact CSN complex is involved in the process, most probably through regulating the level of ubiquitination through deneddylation.

Response: Thanks for your constructive comments. We have added related expressions in the discussion section of the manuscript.

- Line 359 - I suggest to change the word "and" to "or".

Response: Thank you for your careful work. We have modified it in our text.

- Line 361 - I may have missed this information but where was RT-PCR used in this experiment?

Response: We apologize for the inaccurate expression. RT-PCR can be understood as "Reverse-transcription PCR" or "quantitative real-time PCR". We have changed the "RT-PCR" in lines 375 and 669 to "quantitative real-time PCR". As shown in Supplementary Figure 2C, we verified the copy number of the resistance gene hygromycin in the wild-type and mutants by quantitative real-time PCR.

September 30, 2022

Dr. Xiao-Hong Liu
Institute of Biotechnology
Hangzhou
China

Re: Spectrum02020-22R1 (A subunit of the COP9 signalosome, MoCsn6, is involved in fungal development, pathogenicity, and autophagy in rice blast fungus)

Dear Dr. Xiao-Hong Liu:

Thank you for submitting your manuscript to Microbiology Spectrum. As you will see your paper is very close to acceptance. Please modify the manuscript along the lines I have recommended. As these revisions are quite minor, I expect that you should be able to turn in the revised paper in less than 30 days, if not sooner. If your manuscript was reviewed, you will find the reviewers' comments below.

When submitting the revised version of your paper, please provide (1) point-by-point responses to the issues raised by the reviewers as file type "Response to Reviewers," not in your cover letter, and (2) a PDF file that indicates the changes from the original submission (by highlighting or underlining the changes) as file type "Marked Up Manuscript - For Review Only". Please use this link to submit your revised manuscript. Detailed instructions on submitting your revised paper are below.

Link Not Available

Sincerely,

Patricia Albuquerque

Reviewer comments:

Reviewer #1 (Comments for the Author):

The manuscript looks much better now.

Reviewer #2 (Comments for the Author):

Figure S3- Please label the wounded barley and wounded rice leaves in the picture.

How about the appressorium penetration rate of the mutant as compared to the wild type? Could the defect in colonization be due to a defect or delay in penetration?

Reviewer #3 (Comments for the Author):

I thank the authors for their elaborate reply. They answered many of my questions in detail in the response to the reviewers document but did not refer to these points in the manuscript. I suggest to make another effort to clarify as much as possible in the text and make sure that the descriptions and explanations to my questions can be easily understood when reading the manuscript.

In addition:

i. "...We also think that the question of whether the different CSN subunits work individually or in a combined complex is very interesting and worth investigating. Therefore, we tried to knock out the 7 CSN subunits identified in the rice blast fungus. However, only Δ Mocsn6 mutant strain was obtained. Possibly because of the important function of CSN subunits in the rice blast fungus, the genes encoding other subunits failed to be knocked out. Our future research will continue to focus on this issue and try to explore it through other methods." - I suggest to briefly mention this in the discussion.

ii. Lines 115-120 - How do the dual interaction experiments found in Figure 2B agree with the known structure of the CSN complex?

Response: "The amino acid sequences encoding mammalian CSN subunits were used for alignment in the NCBI databases, and the obtained CSN proteins in *M. oryzae* were used for the yeast two-hybrid assay in Figure 2B. Then, the genes encoding the CSN proteins of *M. oryzae* were searched in the UniProt databases to obtain the relevant information of the CSN structure."

This is not what I asked. I meant to ask if the findings in your paper about the pairwise interactions are similar to the published structure of the mammalian CSN. A short indication in the discussion as to the experimental structural similarity would be nice.

iii. Statistical analyses - normalization can overcome differences in overall signal intensity and background between blots. I would make another effort to add statistics.

Preparing Revision Guidelines

Please return the manuscript within 60 days; if you cannot complete the modification within this time period, please contact me. If you do not wish to modify the manuscript and prefer to submit it to another journal, please notify me of your decision immediately so that the manuscript may be formally withdrawn from consideration by Microbiology Spectrum.

Response to reviewer comments

Manuscript Number: Spectrum02020-22R1

Title: A subunit of the COP9 signalosome, MoCsn6, is involved in fungal development, pathogenicity, and autophagy in rice blast fungus

We are thankful to the editors and reviewers for pointing out some important modifications needed in our manuscript. We have thoughtfully taken into account these comments. The following list our responses to the comments on a point-to-point basis.

Reviewer #1 (Comments for the Author):

The manuscript looks much better now.

Response: Thank you for your careful review of our paper.

Reviewer #2 (Comments for the Author):

Figure S3- Please label the wounded barley and wounded rice leaves in the picture.

Response: Thank you for pointing this out, we have added it according to the suggestion.

How about the appressorium penetration rate of the mutant as compared to the wild type?

Response: As shown in Figure 6D, the appressorium penetration rate was significantly reduced in the mutant compared with the wild-type.

Could the defect in colonization be due to a defect or delay in penetration?

Response: Yes. Defects or delays in penetration are responsible for the defect in colonization. When infection rates were counted at 96 hpi (Figure 6D), we observed that there was significantly more uninfected appressorium in the mutant than in the wild-type.

Reviewer #3 (Comments for the Author):

I thank the authors for their elaborate reply. They answered many of my questions in detail in the response to the reviewers document but did not refer to these points in the manuscript. I suggest to make another effort to clarify as much as possible in the text and make sure that the descriptions and explanations to my questions can be easily understood when reading the manuscript.

Response: Thank you for your detailed review of our manuscript. Following your suggestion, we have added more descriptions and explanations to the manuscript.

Line 101: delete “core proteins”, add “core protein MoAtg6”

Line 107-108: add “The “coverage” is a parameter of the sequence alignment result, which means the coverage of the submitted sequence relative to the target sequence.”

Line 260-261: add “The degradation of GFP-MoAtg8 indicates the rate of autophagosome fusion with the vacuole.”

Line 281-282: add “The strain with strong promoter H3 was used to detect the effect of high MoCsn6 expression on the overall ubiquitination level.”

Line 289-281: add “To detect the ubiquitination level of MoAtg6, the MoAtg6-GFP vector was transformed into 70-15 and $\Delta Mocsn6$ mutant. Three of the 70-15 and three of the $\Delta Mocsn6$ mutant expressing MoAtg6-GFP protein were used as three independent replicates.”

Line 464-466: add “13.5% SDS-PAGE with 6 M urea was used to separate Atg8 and Atg8-PE with similar molecular weight. 12.5% SDS-PAGE without additional urea were used to detect the exogenous insertion protein GFP-Atg8/GFP.”

In addition:

i. "...We also think that the question of whether the different CSN subunits work individually or in a combined complex is very interesting and worth investigating. Therefore, we tried to knock out the 7 CSN subunits identified in the rice blast fungus. However, only $\Delta Mocsn6$ mutant strain was obtained. Possibly because of the important function of CSN subunits in the rice blast fungus, the genes encoding other subunits failed to be knocked out. Our future research will continue to focus on this issue and try to explore it through other methods." - I suggest to briefly mention this in the discussion.

Response: Thank you for your suggestion and we have added relevant expressions to the discussion.

Line 386-388: add “Previous studies of CSN have rarely addressed the question of whether and to what extent different CSN subunits work separately or in a complex. We have obtained the deletion mutants for other CSN subunits and will continue to explore this interesting question.”

ii. Lines 115-120 - How do the dual interaction experiments found in Figure 2B agree with the known structure of the CSN complex?

Response: "The amino acid sequences encoding mammalian CSN subunits were used for alignment in the NCBI databases, and the obtained CSN proteins in *M. oryzae* were used for the yeast two-hybrid assay in Figure 2B. Then, the genes encoding the CSN proteins of *M. oryzae* were searched in the UniProt databases to obtain the relevant information of the CSN structure."

This is not what I asked. I meant to ask if the findings in your paper about the pairwise interactions are similar to the published structure of the mammalian CSN. A short indication in the discussion as to the experimental structural similarity would be nice.

Response: Thank you for your suggestion and we have added relevant expressions to the discussion.

Line 328-331: add “In addition, we identified CSN subunits in *M. oryzae* with mammalian homology. The MPN domain and PIC domain in the CSN subunit of *M. oryzae* are corresponding to those of mammals. The experimental results of the interaction between CSN subunits also prove that the COP9

signalosome is highly conservative.”

iii. Statistical analyses - normalization can overcome differences in overall signal intensity and background between blots. I would make another effort to add statistics.

Response: Thanks for your suggestion. We have carried out the statistical analysis of the data and added it to Figure 9.

October 24, 2022

Dr. Xiao-Hong Liu
Institute of Biotechnology
Hangzhou
China

Re: Spectrum02020-22R2 (A subunit of the COP9 signalosome, MoCsn6, is involved in fungal development, pathogenicity, and autophagy in rice blast fungus)

Dear Dr. Xiao-Hong Liu:

Your manuscript has been accepted, and I am forwarding it to the ASM Journals Department for publication. You will be notified when your proofs are ready to be viewed.

Sincerely,

Patricia Albuquerque
Editor, Microbiology Spectrum
